# PerLDiff: Controllable Street View Synthesis Using Perspective-Layout Diffusion Model

## Abstract

Controllable generation is considered a potentially vital approach to address the challenge of annotating 3D data, and the precision of such controllable generation becomes particularly imperative in the context of data production for autonomous driving. Existing methods focus on the integration of diverse generative information into controlling inputs, utilizing frameworks such as GLIGEN or ControlNet, to produce commendable outcomes in controllable generation. However, such approaches intrinsically restrict generation performance to the learning capacities of predefined network architectures. In this paper, we explore the integration of controlling information and introduce PerLDiff (**Per**spective-**L**ayout **Diff**usion Models), a method for effective street view image generation that fully leverages perspective 3D geometric information. Our PerLDiff employs 3D geometric priors to guide the generation of street view images with precise object-level control within the network learning process, resulting in a more robust and controllable output. Moreover, it demonstrates superior controllability compared to alternative layout control methods. Empirical results justify that our PerLDiff markedly enhances the precision of generation on the NuScenes and KITTI datasets.

## 1 Introduction

The advancement of secure autonomous driving systems is fundamentally dependent on the accurate perception of the vehicle's environment. Recently, perception utilizing Bird's Eye View (BEV) has seen rapid progress, markedly pushing forward areas such as 3D object detection (Li et al., 2022; Liu et al., 2023) and BEV segmentation (Zhou & Krähenbühl, 2022). Nonetheless, these systems necessitate extensive datasets with high-quality 3D annotations, the acquisition of which typically involves two consecutive steps: data scene collection and subsequent labeling. Each of these steps incurs significant expenses and presents considerable challenges in terms of data acquisition.

To mitigate issue of data scarcity, the adoption of generative technologies (Van Den Oord et al., 2017; Esser et al., 2021; Ho et al., 2020) has proven practical for reversing the order of data annotation. The paradigm of this approach is to use the collected annotation as controlling information to generate the corresponding lifelike images depicting urban street scenes. By implementing this strategy, it is possible to dramatically lower the costs associated with data annotation while also facilitating the generation of extensive long-tail datasets, subsequently leading to improvements in the perception model's performance. Pioneering research, exemplified by BEVGen (Swerdlow et al., 2024), harnesses the capabilities of autoregressive transformers (Van Den Oord et al., 2017; Esser et al., 2021) to render detailed visualizations of street scenes. Additionally, subsequent studies such as BEVControl (Yang et al., 2023) and MagicDrive (Gao et al., 2023) employ diffusion-based techniques, including GLIGEN (Li et al., 2023b) and ControlNet (Zhang et al., 2023), to integrate controlling information through a basic cross-attention mechanism. However, these methods simply extract integrated conditional features from controlling information to adjust the generation process and are limited in making full use of detailed geometric layout information for accurate attention map manipulation. While these techniques make strides towards meeting the requirements for generating 3D annotations, Fig. 1 demonstrates that there remains significant potential for improvement in scene and object controllability.

To this end, in this paper, we introduce the perspective-layout diffusion models (PerLDiff), a novel method specifically designed to enable precise control over street view image generation at the

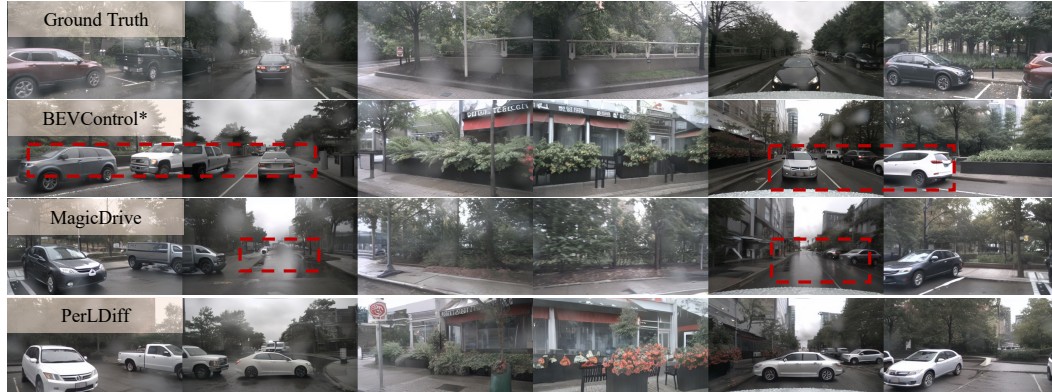

(a) Object Controllability (rotate + 90°)

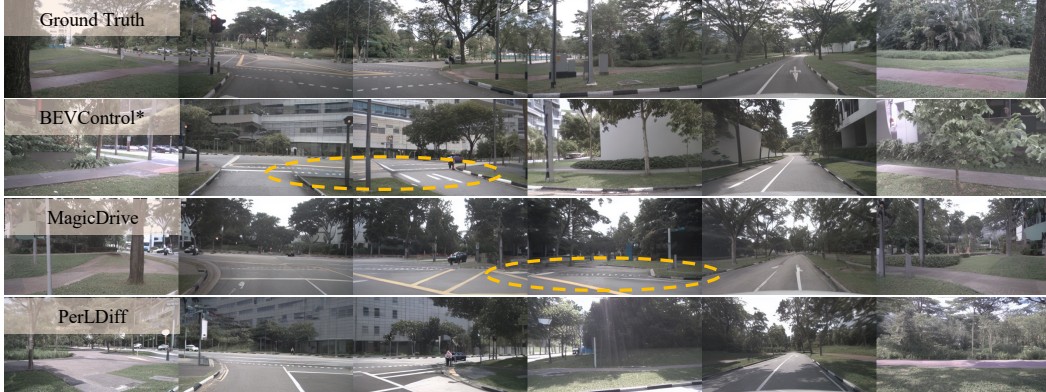

(b) Scene Controllability

Figure 1: PerLDiff enhances controllability over BEVControl* and MagicDrive using geometric priors. *Top*: Demonstrates object controllability by adjusting the 3D annotation yaw by 90 degrees. *Bottom*: Shows scene controllability through the alignment of the street map with the generated image. Regions highlighted by red rectangles and yellow circles indicate areas where the generated images fail to achieve control and alignment with ground truth conditions.

object level. In addition to extracting integrated conditional features from controlling condition information, *i.e.* 3D annotations, our PerLDiff model explicitly renders perspective layout masking maps as geometric priors. Subsequently, a PerL-based controlling module (PerL-CM) is proposed to leverage the geometric priors, *i.e.* perspective layout masking maps. Within PerL-CM, an innovative PerL-based cross-attention mechanism is utilized to accurately guide the generation of each object with their corresponding condition information. We integrate PerL-CM into the pre-trained Stable Diffusion model (Rombach et al., 2022) and fine-tune it on our training dataset. Consequently, our PerLDiff incorporates the formidable generative capabilities of Stable Diffusion with the finely detailed geometric priors of perspective layouts, effectively harnessing their respective strengths for precise object-level image synthesis. Overall, our PerLDiff is capable of generating precise, controllable street view images while also maintaining high fidelity (see Section 3 for details).

The main contributions of this paper are summarized as following three-folds: **(i)** We present PerLDiff, a newly developed framework crafted to synthesize street view images from user-defined 3D annotations. Our PerLDiff carefully orchestrates the image generation process at the object level by leveraging perspective layout masks as geometric priors. **(ii)** We propose a PerL-based cross-attention mechanism that utilizes perspective layout masking maps from 3D annotations to enhance the underlying cross-attention mechanism within PerL-CM. This method enables precise control over the street view image generation process by integrating road geometry and object-specific information derived from 3D annotations. **(iii)** Our PerLDiff method attains state-of-the-art performance on the NuScenes (Caesar et al., 2020) and KITTI (Geiger et al., 2012) dataset compared to existing methods, markedly enhancing detection and segmentation outcomes for synthetic street view images. Furthermore, it holds the potential to function as a robust traffic simulator in the future.

## 2 RELATED WORK

### 2.1 DIFFUSION-BASED GENERATIVE MODELS IN IMAGE SYNTHESIS

Initially developed as a method for modeling data distributions through a sequence of Markov chain diffusion steps (Sohl-Dickstein et al., 2015; Song et al., 2020c;b), diffusion models have undergone rapid advancement. Ho et al. (Ho et al., 2020) introduced denoising diffusion probabilistic models (DDPMs), which have established new benchmarks in the quality of image synthesis. Following efforts have aimed to enhance the efficiency and output diversity of these models by investigating various conditioning strategies (Dhariwal & Nichol, 2021; Choi et al., 2021), architectural adjustments, and training methodologies to refine the image synthesis process (Hong et al., 2023). Nichol and Dhariwal (Dhariwal & Nichol, 2021) proved that diffusion models can be text-conditioned to produce coherent images that are contextually appropriate. Furthermore, advances such as multimodal-conditioned diffusion models have effectively utilized layout images (Li et al., 2023b; Zhang et al., 2023; Rombach et al., 2022; Qu et al., 2023), semantic segmentation maps (Li et al., 2023b; Zhang et al., 2023), object sketches (Voynov et al., 2023; Mou et al., 2023; Li et al., 2023b; Zhang et al., 2023), and depth maps (Mou et al., 2023; Li et al., 2023b; Zhang et al., 2023) to inform the generative process. These methods enable more targeted manipulation of the imagery, thus yielding complex scenes characterized by enhanced structural integrity and contextual pertinence.

### 2.2 DATA GENERATION FOR AUTONOMOUS DRIVING

BEVGen (Swerdlow et al., 2024) represents the pioneering endeavor to harness an autoregressive Transformer (Van Den Oord et al., 2017; Esser et al., 2021) for synthesizing multi-view images pertinent to autonomous driving. Building upon this, BEVControl (Yang et al., 2023) introduces a novel method that incorporates a diffusion model (Ho et al., 2020) for street view image generation, and integrates cross-view attention mechanisms to maintain spatial coherence across neighboring camera views. Subsequently, MagicDrive (Gao et al., 2023) propels the field forward by refining the method for controlling input conditions, drawing insights from ControlNet (Zhang et al., 2023). DrivingDiffusion (Li et al., 2023a) further augments the framework by introducing a consistency loss designed to achieve the perceptual uniformity requisite for high precision in the generation of video from autonomous driving. Panacea (Wen et al., 2023) broadens the capabilities of the model by tackling the challenge of ensuring temporal consistency in video. In contrast to the above approaches, which primarily utilize controlling input conditions to steer the image generation process, our PerLDiff exploits detailed geometric layout information from the input to directly guide object generation with higher precision.

### 2.3 GEOMETRIC CONSTRAINTS IN IMAGE GENERATION

Incorporating geometric priors into image synthesis has been explored to a lesser extent. Work on 3D-aware image generation (Nguyen-Phuoc et al., 2019; Niemeyer & Geiger, 2021) suggests the feasibility of integrating geometric information into generative processes to improve spatial coherence. Nevertheless, these methods often rely on complex 3D representations and may not be directly applied to diffusion model frameworks. Recently, BoxDiff (Xie et al., 2023) reveals a spatial correspondence between the attention map produced by the diffusion model and the corresponding generated image. During the testing phase, the geometric configuration of the attention map's response values is adjusted to yield a more precise image generation. ZestGuide (Couairon et al., 2023) introduces a loss function that enforces a geometric projection onto the attention map, further refining the shape of the attention map's response values to closely approximate the geometric projection of the control information during the inference stage. However, utilizing text prompts to facilitate the generation of complex urban environment layouts poses inherent challenges, owing to the need for crafting intricate prompts to accurately depict urban environments. Furthermore, modifying the cross-attention map to impose strict constraints during the denoising phase of inference can disrupt the intrinsic relationships, leading to a suboptimal approach to synthesizing controllable images. In contrast, our PerLDiff incorporates geometric prerequisites as training priors to guide the generation of street view images, offering a more effective solution.

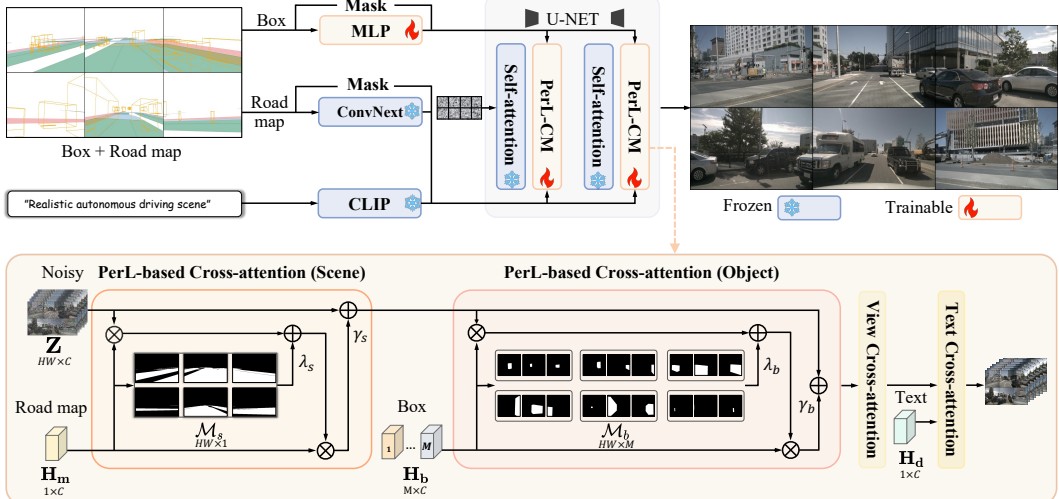

Figure 2: Overview of **PerLDiff** framework for multi-view street image generation. our PerLDiff utilizes perspective layout masking maps derived from 3D annotations to integrate scene information and object bounding boxes. **PerL-CM** is responsible for integrating control information through employing **PerL-based cross-attention (Scene & Object)** mechanism, using PerL masking map (road & box) as geometric priors to guide object-level image generation with high precision. View cross-attention ensures consistency across multiple views, while Text cross-attention integrates textual scene descriptions to facilitate further adjustments.

## 3 CONTROLLABLE STREET VIEW GENERATION BASED ON PERSPECTIVE LAYOUT

In this paper, we introduce PerLDiff, depicted in the Fig. 2, which is designed to enable controllable multi-view street scene generation using 3D annotations. Specifically, PerLDiff leverages perspective projection information from 3D annotations as controlling condition within the training regimen and utilizes perspective layout masks as geometric priors, enabling accurate guidance in object generation. In the following sections, we delineate the process of encoding the controlling condition information from the 3D annotations in Section 3.1. Additionally, we explain how incorporating the perspective layout knowledge ensures scene and object controllability in street view image generation in Section 3.2.

### 3.1 CONTROLLING CONDITIONS ENCODING

Given 3D annotations of a street scene, our goal is to generate multi-view street images. To be more specific, for controllable street view image generation, we extract not only scene information (*i.e.*, textual scene descriptions $\mathbf{S}_d$ and street maps $\mathbf{M}$ revealing features such as road markings and obstacles) but also object information (*i.e.*, bounding box parameters $\mathbf{P}$ and the associated object category $\mathbf{Y}$) from 3D annotations as controlling conditions. These controlling conditions encompass rich semantic and geometric information, so establishing a robust encoding method to utilize this information is essential for generating street view images. Hereafter, we present our controlling condition encoding approach. For simplicity, we omit the details of multi-view perspectives.

**PerL Scene Information** encompasses perspective scene images and supplemental data specific for the whole scene. Typically, the selected scene for annotation is coupled with a street map of the driving environment, which visually differentiates between the road and other background elements using distinct colors. In addition, a generic textual description of the scene is customizable to align with particular scenarios. We employ ConvNext (Liu et al., 2022) and the CLIP text encoder (Radford et al., 2021) to encode the perspective road map, denoted as $\mathbf{S}_m$, derived from the projection of the street map and the textual scene description $\mathbf{S}_d$, respectively. This approach results in the generation of encoded scene features $\mathbf{H}_m \in \mathbb{R}^{1 \times C}$ for the road map and $\mathbf{H}_d \in \mathbb{R}^{1 \times C}$ for the textual scene description:

$$\mathbf{H}_m = \text{ConvNext}(\mathbf{S}_m), \qquad \mathbf{H}_d = \varphi(\mathbf{S}_d). \qquad (1)$$

**PerL Object Information** encapsulates perspective geometric data alongside object category information, which stems from the projections of annotated 3D boxes. This element plays a pivotal role in enabling inverse 3D labeling. Through projecting 3D annotations onto their corresponding perspective images, we ascertain eight 2D corner points for each bounding box within a single image, denoted as $\mathbf{P}_g \in \mathbb{R}^{M \times 2 \times 8}$, where $M$ represents the maximum number of bounding boxes and eight corresponds to the number of corners per bounding box. In conjunction with the object's categorical text $\mathbf{P}_c = \{\mathbf{P}_{c_i}\}_{i=1}^{M}$, we derive the encoded box geometric features $\mathbf{H}_g \in \mathbb{R}^{M \times C}$ and the box categorical features $\mathbf{H}_c \in \mathbb{R}^{M \times C}$, which are illustrated as follows:

$$\mathbf{H}_g = \mathcal{F}(\mathbf{P}_g), \qquad \mathbf{H}_c = \varphi(\mathbf{P}_c), \tag{2}$$

where $\mathcal{F}(\cdot)$ is the Fourier embedding (Mildenhall et al., 2021) function, $\varphi(\cdot)$ represents the pre-trained text embedding encoder of CLIP (Radford et al., 2021) and $C$ representing the dimension of features. Furthermore, we concatenate the encoded geometric features $\mathbf{H}_g$ and categorical features $\mathbf{H}_c$ and subsequently pass the concatenated vector through a Multilayer Perceptron (MLP) (Taud & Mas, 2018) $\mathbf{F}$, to achieve feature fusion $\mathbf{H}_b \in \mathbb{R}^{M \times C}$. The resulting fused box feature representation is given by:

$$\mathbf{H}_b = \mathbf{F}([\mathbf{H}_g, \mathbf{H}_c]). \tag{3}$$

We subsequently input the encoded conditions into the denoising diffusion model to guide the generation process. This is achieved utilizing PerL-based cross-attention mechanism that incorporates PerL masking maps, as detailed below.

### 3.2 Object Controllability via PerL-based Controlling Module

PerL-based Controlling Module (PerL-CM) is responsible for integrating controlling condition information, which encompasses both the PerL scene and object information, into the latent feature maps of noisy street view images. This integration is primarily achieved via the scene and object PerL-based cross-attention mechanism. Initially, this mechanism assigns initial values to the attention maps, under the guidance of road and box PerL masking maps. Throughout the training of the network, these values are optimized to ensure that the response of the attention map accurately corresponds to the regions where the objects are located. Subsequently, the information from both the road map and the bounding box are sequentially integrated into the noise street view image. To more effectively integrate PerL scene and object information, a gating operation is used, similar to the method employed in GLIGEN (Li et al., 2023b), which dynamically adjusts the contribution of condition information according to the adaptive process. To ensure multi-view consistency, View Cross-attention leverages information from the immediate left and right views for uniformity across various perspectives. Additionally, Text Cross-attention manipulates the weather and lighting conditions of street scenes using textual scene description.

**PerL Masking Map (road & box)** is comprised of PerL road masking map $\mathcal{M}_s \in \mathbb{R}^{HW \times 1}$ and PerL box masking map $\mathcal{M}_b \in \mathbb{R}^{HW \times M}$, where $H$ and $W$ represent the height and width dimensions of the image, respectively. These masking maps are articulated as follows:

$$\mathcal{M}_s = \Upsilon(\mathbf{S}_m), \quad \mathcal{M}_b = \Phi(\mathbf{P}_g), \tag{4}$$

where $\Upsilon(\cdot)$ generates the masking map for the non-empty regions of the projected road maps. Meanwhile, $\Phi(\cdot)$ produces the masking map corresponding to the inner region of each projected 3D bounding box for every perspective image, enabling precise control at the object level.

**PerL-based Cross-attention (Scene & Object)** leverages the prior masking maps to enhance the learning of cross-attention between the input controlling conditions and the noisy street view images. As depicted in Fig. 3, the cross-attention map exhibits perceptual equivalence with the generated street view image. However, this correspondence is imprecise and lacks the necessary alignment during the training stage. To this end, our approach utilizes a PerL-based cross-attention mechanism that incorporates geometric knowledge derived from both the scene context and the bounding box into the computation of the cross-attention map. In PerLDiff, the road map and object bounding box data are seamlessly merged with the noisy street view images throughout each stage of the denoising process. For the sake of notational simplicity, the linear embeddings and normalization steps typically involved in the attention mechanism have been omitted.

$$\mathcal{A}_s = softmax(\lambda_s \cdot \mathcal{M}_s + \mathbf{Z}\mathbf{H}_m^T / \sqrt{d}), \quad \mathcal{A}_b = softmax(\lambda_b \cdot \mathcal{M}_b + \mathbf{Z_s}\mathbf{H}_b^T / \sqrt{d}) \tag{5}$$

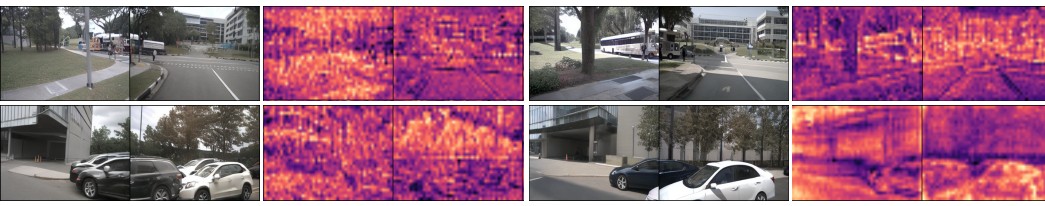

BEVControl*                                                              PerLDiff (Ours)

Figure 3: Visualization of cross-attention maps reveals perceptual congruency with the generated image. BEVControl* produces disorganized and vague attention maps, which result in inferior image quality. Conversely, our PerLDiff method fine-tunes the response within the attention maps, resulting in more accurate control information at the object level and improved image quality. See more qualitative examples in the Appendix D.

where $\lambda_s$ and $\lambda_b$ are weight parameters that control the influence of the masking map, $d$ denotes the dimensionality, and $\mathbf{Z}$, $\mathbf{Z}_b$, $\mathbf{Z}_s \in \mathbb{R}^{HW \times C}$ represent different noisy street view images. Here, $\mathcal{A}_s \in \mathbb{R}^{HW \times 1}$ characterizes the association between the road map and the noisy image, while $\mathcal{A}_b \in \mathbb{R}^{HW \times M}$ clarifies the relationship between the conditions of the object bounding box and the noisy image. For visualization purposes, as shown in Fig. 3, we average $\mathcal{A}_b$ along the second dimension and merge all object attention maps into one representation at step 50 of the DDIM process in the last block of UNet. The final noisy street view image is synthesized through an attention-based aggregation mechanism enhanced by a residual connection, which can be expressed as:

$$\mathbf{Z_s} = \gamma_s \cdot \mathcal{A}_s \mathbf{H}_m + \mathbf{Z}, \qquad \mathbf{Z_b} = \gamma_b \cdot \mathcal{A}_b \mathbf{H}_b + \mathbf{Z_s}, \qquad (6)$$

where $\gamma_s$ and $\gamma_b$ represent learnable parameters modulating the influence of respective conditions.

**View Cross-attention** is corroborated by preliminary works such as BEVControl (Yang et al., 2023), MagicDrive (Gao et al., 2023), DrivingDiffusion (Li et al., 2023a), and Panacea (Wen et al., 2023), plays an instrumental role in facilitating the synthesis of images that maintain visual consistency across varying camera perspectives. For additional information, please see the Appendix C.3.

**Text Cross-attention** enhances the Stable Diffusion (Rombach et al., 2022) model's ability to modulate street scenes through textual scene description. This capability is crucial for dynamically adapting the rendering of street scenes to accommodate various lighting and weather conditions. By integrating detailed textual scene description, our PerLDiff can effectively alter visual elements such as illumination and atmospheric effects, ensuring that the generated images reflect the specified conditions accurately. Please refer to Fig. 4 for qualitative examples of this enhancement.

### 3.3 DISCUSSION

In contrast to previous approaches for autonomous driving such as BEVControl (Yang et al., 2023), MagicDrive (Gao et al., 2023), DrivingDiffusion (Li et al., 2023a), and Panacea (Wen et al., 2023), which employ a basic cross-attention mechanism to integrate controlling condition information, our PerLDiff leverages geometric priors via PerL masking map. This approach directs the generation of each object with its respective control information during the training phase, effectively countering the common misalignment between the attention map and condition information that often results in compromised image controllability. For instance, the attention map of BEVControl (Yang et al., 2023), illustrated in Fig. 3, demonstrates disorganized patterns and lacks precision. Conversely, our PerLDiff markedly enhances the accuracy of generated images and the granularity of condition information at the object level by ensuring meticulous guidance within the attention map. For more qualitative results, please see the Appendix D.

### 4 EXPERIMENTS

We assess PerLDiff's ability to control quality across multiple benchmarks, including multi-view 3D object detection, BEV segmentation and monocular 3D object detection. Subsequently, we conduct ablation studies to ascertain the contribution of each component within our proposed methodology.

Table 1: Comparison of the controllability of street view image generation on NuScenes (Caesar et al., 2020) *validation* set. Our replication of BEVControl*, serving as the baseline, employs standard cross-attention mechanisms contrary to PerL-based cross-attention utilized in our PerLDiff. Outcomes demonstrating superior performance are highlighted in **bold**.

| Method | Detector | FID↓ | mAP↑ | NDS↑ | mAOE↓ | Road mIoU↑ | Vehicle mIoU↑ |
|---|---|---|---|---|---|---|---|
| Oracle | BEVFormer | – | 27.06 | 41.89 | 0.54 | 70.35 | 33.36 |
| Oracle | BEVFusion | – | 35.54 | 41.20 | 0.56 | 70.46 | 35.86 |
| MagicDrive (Gao et al., 2023) | BEVFusion | 16.20 | 12.30 | 23.32 | – | 61.05 | 27.01 |
| BEVControl* | BEVFusion | **13.05** | 9.98 | 19.61 | 0.94 | 60.74 | 22.47 |
| PerLDiff (Ours) | BEVFusion | 13.36 | **15.24** | **24.05** | **0.78** | **61.26** | **27.13** |
| BEVGen (Swerdlow et al., 2024) | – | 25.54 | – | – | – | 50.20 | 5.89 |
| BEVControl (Yang et al., 2023) | BEVFormer | 24.85 | 19.64 | 28.68 | 0.78 | 60.80 | 26.86 |
| BEVControl* | BEVFormer | 13.05 | 16.48 | 28.08 | 0.88 | 60.74 | 22.47 |
| PerLDiff (Ours) | BEVFormer | 13.36 | **25.10** | **36.24** | **0.72** | **61.26** | **27.13** |

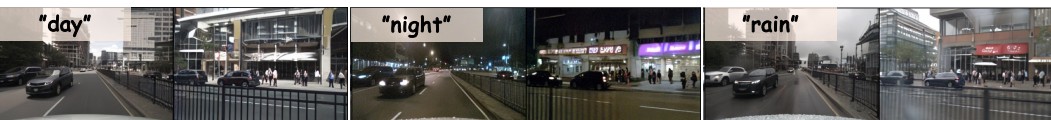

Figure 4: Qualitative visualization on NuScenes demonstrating the effects of Text Cross-attention. From left to right: *day*, *night*, and *rain* scenarios synthesized by PerLDiff, highlighting its adaptability to different lighting and weather conditions. For more examples, see the Appendix D.

### 4.1 DATASETS

**NuScenes** dataset comprises 1,000 urban street scenes, traditionally segmented into 700 for training, 150 for validation, and 150 for testing. Each scene features six high-resolution images (900×1600), which together provide a complete 360-degree panoramic view of the surroundings. Additionally, NuScenes includes comprehensive road maps of the driving environment, featuring details such as lane markings and obstacles. We extend the class and road type annotations similar to Magic-Drive (Gao et al., 2023) and NuScenes, incorporating ten object classes and eight road types for map rendering. To address the resolution limitations of the U-Net architecture in Stable Diffusion (Ronneberger et al., 2015), we adopt image resolutions of 256×384 as in BEVFormer (Li et al., 2022), and 256×704 following BEVFusion (Liu et al., 2023).

**KITTI** dataset contains 3,712 images for training and 3,769 images for validation. KITTI dataset has only one perspective image and does not have road map information. Given the varied image resolutions in KITTI (approximately 375×1242), we pad them to 384×1280 for generative learning.

### 4.2 MAIN RESULTS

In this subsection, we assess our PerLDiff's generative quality through the perception results of several pre-trained methods: BEVFormer (Li et al., 2022), BEVFusion (camera-only) (Liu et al., 2023), and StreamPETR (Wang et al., 2023a) for multi-view 3D detection; CVT (Zhou & Krähenbühl, 2022) for BEV segmentation; all trained on the NuScenes set, and MonoFlex (Zhang et al., 2021) for monocular 3D detection trained on the KITTI set. Additionally, we leverage our synthesized dataset to enhance the performance of various 3D detection models (*i.e.*, BEVFormer and Stream-PETR) on the NuScenes *test* set, validating the effectiveness of our PerLDiff.

**Controllable Generation on NuScenes.** To evaluate the effectiveness of PerLDiff, we trained the model on the NuScenes *train* set and subsequently generated a synthetic *validation* set using the provided road maps and 3D annotations. The controllability of PerLDiff was examined by applying perception models, originally trained on the real *train* set, to our synthetic *validation* set. As summarized in Tab. 1, PerLDiff outperforms competing methods across most metrics, as tested with BEVFormer (Li et al., 2022) and BEVFusion (Liu et al., 2023). In a rigorous comparison, we replicated BEVControl (Yang et al., 2023) using identical settings, with the exception of our innovative

Table 2: Controllability comparison on KITTI (Geiger et al., 2012) *validation* set, showcasing vehicle mAP obtained by MonoFlex (Zhang et al., 2021) using data generated by our PerLDiff and the baseline BEVControl*. "NuScenes → KITTI" denotes initial training on NuScenes *train* set followed by fine-tuning on KITTI *train* set.

| Method | KITTI | | | | NuScenes → KITTI | | | |
|---|---|---|---|---|---|---|---|---|
| | Easy↑ | Mod.↑ | Hard↑ | FID↓ | Easy↑ | Mod.↑ | Hard↑ | FID↓ |
| Oracle | 22.29 | 15.54 | 13.38 | – | 22.29 | 15.54 | 13.38 | – |
| BEVControl* | 0.33 | 0.29 | 0.39 | 39.47 | 1.32 | 1.51 | 1.64 | 32.96 |
| PerLDiff (Ours) | **11.04** | **7.44** | **6.03** | **39.03** | **13.12** | **9.24** | **7.59** | **31.70** |

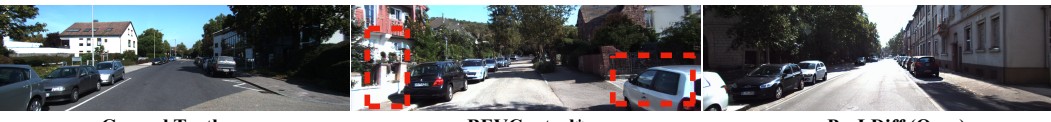

**Ground Truth**   **BEVControl***   **PerLDiff (Ours)**

Figure 5: Qualitative visualization comparison on KITTI (Geiger et al., 2012). Red markers denote instances where BEVControl* inaccurately generates output compared to PerLDiff and ground truth.

element: the PerL-based cross-attention mechanism. PerLDiff demonstrates notable improvements, with increases of 8.62%, 8.16%, and 0.16% in mean Average Precision (mAP), NuScenes Detection Score (NDS), and mean Average Orientation Error (mAOE), respectively, compared to BEVControl* when using BEVFormer. With BEVFusion, it achieves gains of 5.26%, 4.44%, and 0.16% in these metrics against BEVControl*, confirming its effectiveness at a resolution of $256 \times 384$. The superiority of PerLDiff is further affirmed by BEV segmentation metrics (Zhou & Krähenbühl, 2022), significantly outperforming BEVControl* with a 0.52% increase in Road mIoU and a 4.66% increase in Vehicle mIoU, validating the efficacy of the PerL-based cross-attention mechanism in enhancing scene controllability. Regarding the Fréchet Inception Distance (FID) (Heusel et al., 2017) metric, our results are comparable to those of BEVControl*. While PerLDiff incorporates prior constraints to ensure accuracy in object detection, this may adversely affect the details in the background of the images. As illustrated in Fig. 14 of Appendix D, PerLDiff produces background details that do not align with those of real images.

Compared to the state-of-the-art MagicDrive (Gao et al., 2023), our method demonstrates superior performance across all metrics, particularly in the FID metric, reflecting an improvement of 2.84%. Additionally, we achieve a 2.94% improvement in mAP and a 0.73% increase in NDS. These results substantiate the strengths of PerLDiff in terms of both generation quality and controllability.

**Controllable Generation on KITTI.** The scarcity of training data in the KITTI dataset (Geiger et al., 2012) often limits a generative model's ability to understand the relationship between control information and image synthesis. To address this challenge, we implement two distinct strategies for generating images within the KITTI framework: one strategy involves direct training using the KITTI *train* set, while the other entails initial training on the NuScenes *train* set followed by fine-tuning on the KITTI *train* set. In Tab. 2, we present the results of monocular 3D object detection on the KITTI *validation* set, utilizing a pretrained MonoFlex (Zhang et al., 2021) detector. As illustrated in Fig. 5, the naive approach results in significant misalignment between the labels and the corresponding generated images, leading to a considerable performance gap: 11.04 vs. 0.33 for one metric and 13.12 vs. 1.32 for another. There are two main reasons for the observed differences. First, the limited size of the KITTI dataset, which contains just 3,712 training images, impedes the learning process of traditional methods that do not utilize PerL masking map. Second, monocular 3D object detection is highly sensitive to accurate depth prediction. Depth is derived from the 2D projected size and the estimated 3D size through perspective projection. Our method produces more precise object sizing, thereby enhancing detection performance. More visual results on KITTI can be found in the Appendix D.

**Boosting Perception Models Using Synthesized Dataset.** Generative models have become widely acknowledged as effective tools for data augmentation, thereby improving the generalization capabilities of perception models. To evaluate this approach, we leverage our synthesized dataset to improve the performance of various detection models on NuScenes *test* set. The gains presented in the second row of Tab. 3 confirm that augmenting with data annotated optimally (*i.e.*, using the

Table 3: Performance comparison for the boosting performance of 3D detection models using synthesized dataset on NuScenes (Caesar et al., 2020) *test* set using BEVFormer (Li et al., 2022) and StreamPETR (Wang et al., 2023a). The "*train* + Real *val*" configuration serves as a benchmark, representing the ideal upper performance limit achievable. "Syn. *val*\*" represents the synthetic *validation* set generated by BEVControl. The numbers in parentheses indicate the performance disparity relative to the "*train* + Real *val*" configuration.

| Training | Detector | mAP↑ | NDS↑ | mATE↓ | mASE↓ | mAOE↓ |
|---|---|---|---|---|---|---|
| *train* | | 28.97 | 42.52 | 72.90 | 28.15 | 56.34 |
| *train* + Real *val* | BEVFormer | 32.20 | 45.44 | 69.43 | 27.40 | 52.88 |
| *train* + Syn. *val*\* | | 29.92 (-2.28%) | 43.20 (–2.24%) | 70.76 (+1.33%) | 27.69 (+0.29%) | 57.57 (+4.69%) |
| *train* + Syn. *val* (Ours) | | **31.66** (-0.54%) | **44.91** (-0.53%) | **70.09** (+0.66%) | **27.56** (+0.16%) | **55.05** (+2.17%) |
| *train* | | 47.84 | 56.66 | 55.91 | 25.81 | 47.40 |
| *train* + Real *val* | StreamPETR | 50.92 | 58.68 | 54.36 | 25.12 | 45.36 |
| *train* + Syn. *val*\* | | 47.37 (-3.55%) | 56.40 (-2.28%) | 56.99 (+2.63%) | 25.58 (+0.46%) | 47.69 (+2.33%) |
| *train* + Syn. *val* (Ours) | | **49.07** (-1.85%) | **57.92** (-0.76%) | **55.71** (+1.35%) | **25.57** (+0.45%) | **47.08** (+1.72%) |

combined real NuScenes *train* + Real *val* set) is beneficial. In particular, the performance of BEVFormer (Li et al., 2022) and StreamPETR (Wang et al., 2023a) improved significantly after the dataset was augmented with real *validation* set. The most notable gains for BEVFormer were observed in the mAP and NDS metrics, which increased by 3.23% and 2.92%, respectively. Similarly, for StreamPETR, increases in mAP and NDS were recorded at 3.08% and 2.02%, respectively.

Furthermore, augmentations using synthetic *validation* set yielded competitive improvements that almost matched the performance gains observed with real *validation* set. The gaps in performance metrics, such as mAP, NDS, and mAOE, were minimal, thus solidifying the value of synthetic augmentation compared to the *train-only* baseline. Specifically, BEVFormer and StreamPETR exhibited only slight gaps in mAP (0.54% and 1.85%), NDS (0.53% and 0.76%) and mAOE (2.17% and 1.72%), respectively. In addition, these discrepancies were even less pronounced compared to BEVControl\*, highlighting the effectiveness of the PerL-based cross-attention mechanism.

## 4.3 ABLATION STUDY

To determine the effectiveness of the fundamental components within our PerLDiff, we perform ablation studies concentrated on key elements: PerL-based cross-attention.

**Effectiveness of PerL-based Cross-attention.** To illustrate the impact of PerL-based cross-attention, we devised a comprehensive comparative experiment, the results of which are presented in Tab. 4. Method (a) employs road map and 3D box as conditions, which are integrated into the model using standard cross-attention (Yang et al., 2023; Gao et al., 2023) with the configuration mirroring that of BEVControl\*. "Box Mask" and "Road Mask" denote the process wherein the control information is merged with the model through PerL-based cross-attention. Method (a)→(b) signifies the adoption of PerL-based cross-attention for the road map, leading to improvements of 0.44% in Road mIoU and 0.32% in NDS. These gains underscore the augmented controllability achieved by combining the road map with PerL-based cross-attention and its efficacy in aligning generated data with real observations. Additionally, Method (a)→(c) results in marked improvements of 9.59% in mAP, 7.99% in NDS, 0.14% in mAOE and 3.80% in Vehicle mIoU, strongly supporting the utility of PerL-based cross-attention in producing accurate data-annotation alignments for objects. To optimally regulate elements of the background and foreground, Method (c), in contrast to baseline Method (a), indicates increases of 8.62% in mAP, 8.16% in NDS, 0.16% in mAOE, 0.52% in Road mIoU and 4.66% in Vehicle mIoU. These results validate the efficiency of PerL-based cross-attention in enhancing image controllability.

**Effectiveness of Masking Map Weight Coefficients.** Tab. 5 examines the effects of varying the masking map weight coefficients $\lambda_s$ and $\lambda_b$, where higher values indicate a greater integration of PerL knowledge into network learning. The table demonstrates that detection metrics improve with increasing values of $\lambda_s$ and $\lambda_b$ within a certain range. However, the FID score also increases, underscoring the significant role of PerL knowledge in the controllable learning process of the diffusion model. For optimal controllability, we set the default values of $\lambda_s$ and $\lambda_b$ to 5.0 in the main text.

**Qualitative Visualization.** We assess the efficacy of our PerLDiff method against the baseline approaches BEVControl\* and MagicDrive, with all methods aimed at synthesizing perspective scene

Table 4: Ablation of the PerL-based cross-attention, reporting 3D object detection improvements using BEVFormer (Li et al., 2022) and BEV segmentation enhancements using CVT (Zhou & Krähenbühl, 2022). Numbers in parentheses indicate performance gains over the baseline.

| Method | Road Mask | Box Mask | FID↓ | mAP↑ | NDS↑ | mAOE↓ | Road mIoU↑ | Vehicle mIoU↑ |
|---|---|---|---|---|---|---|---|---|
| Oracle | | | - | 27.06 | 41.89 | 0.54 | 70.35 | 33.36 |
| (a) | | | **13.05** | 16.48 | 28.08 | 0.88 | 60.74 | 22.47 |
| (b) | ✓ | | 13.20 | 16.27 | 28.40 | 0.86 | 61.18 | 23.14 |
| (c) | | ✓ | 13.54 | **26.07** | 36.07 | 0.74 | 61.21 | 26.27 |
| (d) | ✓ | ✓ | 13.36 | 25.10 (+8.62%) | **36.24** (+8.16%) | **0.72** (-0.16%) | **61.26** (+0.52%) | **27.13** (+4.66%) |

Table 5: Ablation of different values of masking map weight coefficients $\lambda_s$ and $\lambda_b$. We report the 3D object detection results based on BEVFormer (Li et al., 2022) and BEV Segmentation results based on CVT (Zhou & Krähenbühl, 2022).

| Method | $\lambda_s$ $\lambda_b$ | FID↓ | mAP↑ | NDS↑ | mAOE↓ | Road mIoU↑ | Vehicle mIoU↑ |
|---|---|---|---|---|---|---|---|
| Oracle | – | – | 27.06 | 41.89 | 0.54 | 70.35 | 33.36 |
| (a) | 1.0 1.0 | **12.87** | 22.30 | 34.08 | 0.73 | 61.31 | 25.03 |
| (b) | 3.0 3.0 | 14.03 | 24.41 | 35.75 | 0.74 | 60.58 | 26.82 |
| (c) | 5.0 5.0 | 13.36 | **25.10** | **36.24** | **0.72** | 61.26 | **27.13** |
| (d) | 10.0 10.0 | 14.24 | 24.98 | 35.52 | 0.76 | **61.75** | 26.62 |

images. As illustrated in Fig. 1, PerLDiff generates images of substantially superior quality compared to BEVControl* and MagicDrive, particularly in accurately depicting scene controllability and object controllability. More results can be found in Appendix D.

## 5 CONCLUSION

In conclusion, our PerLDiff introduces a streamlined framework that adeptly merges geometric constraints with synthetic street view image generation, harnessing diffusion models' power for high-fidelity visuals. The architecture boasts a PerL-based controlling module (PerL-CM) that, through training, becomes seamlessly integrated with Stable Diffusion. Meanwhile, a cutting-edge PerL-based cross-attention mechanism guarantees meticulous feature guidance at the object level for precise control. Experiments on NuScenes (Caesar et al., 2020) and KITTI (Geiger et al., 2012) datasets confirm our PerLDiff's enhanced performance in image synthesis and downstream tasks like 3D object detection and segmentation. Flexible yet precise, our PerLDiff's method of PerL-based cross-attention with geometric perspective projections during training finely balances image realism with accurate condition alignment.

**Limitation and Future Work.** Fig. 6 depicts several failure cases of our PerLDiff, where the model erroneously generates vehicles with the front and rear orientations reversed, in contrast to the ground truth. This limitation arises from the usage of a PerL mask in our PerLDiff, which does not account for the orientation on the 2D PerL plane. Future endeavors may explore video generation, extending to work such as DrivingDiffusion (Li et al., 2023a), Panacea (Wen et al., 2023), and Driving into the Future (Wang et al., 2023b).

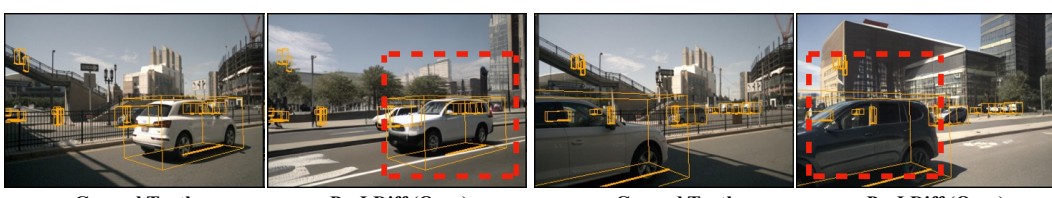

| **Ground Truth** | **PerLDiff (Ours)** | **Ground Truth** | **PerLDiff (Ours)** |

Figure 6: Failure cases of our PerLDiff, with red markers highlighting instances where, compared to the ground truth, our PerLDiff generates images with the front and rear of vehicles reversed.

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

The supplementary material is organized into the following sections:

- Section A: DDPM Preliminaries
- Section B: Implementation Details
- Section C: Additional Experiments
- Section D: Visualization Results

## A  DDPM PRELIMINARIES

Denoising Diffusion Probabilistic Models (DDPM) (Ho et al., 2020) are a class of generation models which simulate a Markov chain of diffusion steps to gradually convert data samples into pure noise. The generative process is then reversed to synthesize new samples from random noise. We commence with an observation $x_0$ sampled from the data's true distribution $q(x)$, and then progressively apply Gaussian noise over a series of $T$ time steps. The forward diffusion is mathematically defined as $q(x_t|x_{t-1}) = \mathcal{N}(x_t; \sqrt{1 - \beta_t}x_{t-1}, \beta_t\mathbf{I})$,, where $\beta_t$ is a variance term that can be either time-dependent or learned during training. The entire forward diffusion process can be represented as the product of the conditional distributions from each step:

$$q(x_{1:T}|x_0) = \prod_{t=1}^{T} q(x_t|x_{t-1}), \tag{7}$$

where the sequence $\{\beta_t\}_{t=1}^{T}$ specifies the noise schedule applied at each timestep. The diffusion process is notable for permitting direct sampling of $x_t$ from $x_0$ using a closed-form expression:

$$q(x_t|x_0) = \sqrt{\bar{\alpha}_t}x_0 + \sqrt{1 - \bar{\alpha}_t}\epsilon, \quad \text{where} \quad \epsilon \sim \mathcal{N}(0, \mathbf{I}), \tag{8}$$

in which $\alpha_t = 1 - \beta_t$ and the cumulative product $\bar{\alpha}_t = \prod_{s=1}^{t} \alpha_s$. To synthesize new samples, a reverse process known as the backward diffusion is learned, which conceptually undoes the forward diffusion. This inverse transition is captured through a parameterized Gaussian distribution:

$$p_\theta(x_{t-1}|x_t) = \mathcal{N}(x_{t-1}; \mu_\theta(x_t), \sigma_\theta^2(x_t)\mathbf{I}). \tag{9}$$

## B  IMPLEMENTATION DETAILS

Our PerLDiff utilizes the pre-trained Stable Diffusion v1.4 (Rombach et al., 2022), augmented with specific modifications to enhance scene control. Training was conducted on a server equipped with eight Tesla V100 (32 GB) GPUs over 60,000 iterations, which required two days. An initial batch size of 16 was adjusted to a per-GPU batch of two for focused optimization, particularly for data samples comprising six view images per frame. The generation of samples conforms to the CFG rule (Ho & Salimans, 2022), employing a guidance scale of 5.0 and the Denoising Diffusion Implicit Models (DDIM) (Song et al., 2020a) across 50 steps.

For scene manipulation, the text encoder within Stable Diffusion is retained, along with a weight-frozen CLIP to manage textual inputs and ConvNext for processing road maps. Feature extraction from PerL boxes is conducted via an MLP, optimized through PerL-based controlling module (PerL-CM) with randomly initialized weights. In contrast, certain modules inherit and freeze pre-trained weights from Stable Diffusion v1.4. The key parameters within PerL-CM, $\lambda_b$ and $\lambda_s$, are set to 5.0 to facilitate optimal image synthesis. Furthermore, DDIM (Song et al., 2020a) and CFG (Ho & Salimans, 2022) are integrated into our training regimen, with a novel approach of omitting all conditions at a rate 10% to foster model versatility.

The optimization process employs AdamW (Loshchilov & Hutter, 2017) without a weight decay coefficient and with a learning rate of $5 \times 10^{-5}$, complemented by a warm-up strategy during the first 1,000 iterations. BEVFormer (Li et al., 2022), StreamPETR (Wang et al., 2023a), and CVT (Zhou & Krähenbühl, 2022) were retrained using original configurations tailored to our target resolution. The performance of BEVFusion (Liu et al., 2023) and MonoFlex (Zhang et al., 2021) was assessed using their provided code and pre-trained weights.

# C ADDITIONAL EXPERIMENTS

In this section, we present additional experiments conducted to validate controllability at different resolutions (256× 704) and to assess the contributions of individual components within our PerLDiff. Our studies focus on the following aspects:

- Effectiveness of Controllable Generation on NuScenes (Subsection C.1)
- Effectiveness of Classifier-Free Guidance Scale (Subsection C.2)
- Effectiveness of View Cross-attention for Multi-View Consistency (Subsection C.3)
- Effectiveness of Perl-based Cross Attention (Object) (Subsection C.4)
- Effectiveness of Multiplication in Perl-based Cross Attention (Subsection C.5)
- Effectiveness of PerLDiff Based on ControlNet (Subsection C.6)

Our results confirm the superior performance of our method across various resolutions and illustrate how each component is integral to the success of our PerLDiff.

## C.1 EFFECTIVENESS OF CONTROLLABLE GENERATION ON NUSCENES

In Tab. 6, we conduct a comparative analysis to emphasize the capabilities of PerLDiff for controllable generation at a resolution of 256×704. This quantitative evaluation contrasts our method with other leading approaches based on the detection metrics provided by BEVFusion (Liu et al., 2023). Our PerLDiff exhibits significantly superior performance, achieving mAP improvements of 3.84% and 11.50%, and NDS increases of 0.45% and 10.80%, compared to MagicDrive (Gao et al., 2023) and BEVControl*, respectively. These results confirm the efficacy of PerLDiff in the precise controllable generation at the object level.

Table 6: Controllability comparison for street view image generation on the NuScenes *validation* set. A quantitative evaluation using 3D object detection metrics from BEVFusion (Liu et al., 2023).

| Method | FID↓ | mAP↑ | NDS↑ | mATE↓ | mASE↓ | mAOE↓ |
|---|---|---|---|---|---|---|
| Oracle | – | 35.54 | 41.20 | 0.67 | 0.27 | 0.56 |
| MagicDrive (Gao et al., 2023) | 16.59 | 20.85 | 30.26 | – | – | – |
| BEVControl* | 15.94 | 13.19 | 19.91 | 0.94 | 0.34 | 0.96 |
| PerLDiff (Ours) | **15.67** | **24.69** | **30.71** | **0.82** | **0.28** | **0.76** |

## C.2 EFFECTIVENESS OF CLASSIFIER-FREE GUIDANCE SCALE

In Tab. 7, we assess the effect of the CFG (Ho & Salimans, 2022) scale on the sampling of data generation. The term "scale" refers to the CFG scale, which is adjusted to balance conditional and unconditional generation. The transition from Method (b) to (e) indicates an increase in the CFG scale from 5.0 to 12.5. The results show an average increase of 2.87 in FID, an average decrease of 0.87% in mAP, an average reduction of 1.03% in NDS, a 0.02% increase in mAOE and a 1.07% drop in Vehicle mIoU. This provides substantial evidence that an excessively large CFG scale can degrade the quality of generated images and adversely affect various performance metrics.

Table 7: Comparison of different CFG (Ho & Salimans, 2022) scale to each metric. We report the 3D object detection results based on BEVFormer (Li et al., 2022) and BEV Segmentation results based on CVT (Zhou & Krähenbühl, 2022).

| Method | scale | FID↓ | mAP↑ | NDS↑ | mAOE↓ | Road mIoU↑ | Vehicle mIoU↑ |
|---|---|---|---|---|---|---|---|
| Oracle | – | – | 27.06 | 41.89 | 0.54 | 70.35 | 33.36 |
| (a) | 2.5 | **12.36** | 23.89 | 36.03 | **0.70** | 60.05 | 26.95 |
| (b) | 5.0 | 13.36 | **25.10** | **36.24** | 0.72 | 61.26 | **27.13** |
| (c) | 7.5 | 15.52 | 24.62 | 35.60 | 0.74 | **61.52** | 26.63 |
| (d) | 10.0 | 16.32 | 24.20 | 35.05 | 0.73 | 61.43 | 26.00 |
| (e) | 12.5 | 16.86 | 23.86 | 34.98 | 0.74 | 61.25 | 25.55 |

---

**Algorithm 1** PerL-based Controlling Module (PerL-CM)

---

**Input:** road map features $\mathbf{H}_m \in \mathbb{R}^{1 \times C}$, road masking map $\mathcal{M}_s \in \mathbb{R}^{HW \times 1}$, box features $\mathbf{H}_b \in \mathbb{R}^{M \times C}$, box masking map $\mathcal{M}_b \in \mathbb{R}^{HW \times M}$, scene text description features $\mathbf{H}_d \in \mathbb{R}^{1 \times C}$, noisy multi-view street image feature $\mathbf{Z} \in \mathbb{R}^{HW \times C}$, and dimension $d$ (omit the detail of multi-view perspectives)
**Output:** Updated $\mathbf{Z}$
1: $\mathcal{A}_s \leftarrow softmax(\lambda_s \cdot \mathcal{M}_s + \mathbf{Z}\mathbf{H}_m^T/\sqrt{d})$
   *// compute attention map for the road map in PerL-based cross-attention (scene)*
2: $\mathbf{Z_s} \leftarrow \gamma_s \cdot \mathcal{A}_s\mathbf{H}_m + \mathbf{Z}$
3: $\mathcal{A}_b \leftarrow softmax(\lambda_b \cdot \mathcal{M}_b + \mathbf{Z_s}\mathbf{H}_b^T/\sqrt{d})$
   *// compute attention map for the box in PerL-based cross-attention (object)*
4: $\mathbf{Z_b} \leftarrow \gamma_b \cdot \mathcal{A}_b\mathbf{H}_b + \mathbf{Z_s}$
5: $\hat{\mathbf{Z}} \leftarrow \mathbf{Z}_b + \mathcal{C}(\mathbf{Z}_b, \mathbf{Z}_l, \mathbf{Z}_l) + \mathcal{C}(\mathbf{Z}_b, \mathbf{Z}_r, \mathbf{Z}_r)$
   *// maintain visual consistency via View cross-attention*
6: $\mathbf{Z}^* \leftarrow softmax(\hat{\mathbf{Z}}\mathbf{H}_d^T/\sqrt{d})\mathbf{H}_d + \hat{\mathbf{Z}}$
   *// alter illumination and atmospheric effects by Text cross-attention*

---

## C.3 EFFECTIVENESS OF VIEW CROSS-ATTENTION FOR MULTI-VIEW CONSISTENCY

View cross-attention ensures the seamless integration of visual data by maintaining continuity and consistency across the multiple camera feeds that are integral to current multi-functional perception systems in autonomous vehicles. Typically, autonomous vehicles feature a 360-degree horizontal surround view from a BEV perspective, resulting in overlapping fields of vision between adjacent cameras. Consequently, we facilitate direct interaction between the noise maps of each camera and those of the immediate left and right cameras. Given the noisy images from the current, left, and right cameras, designated as $\mathbf{Z}_b$, $\mathbf{Z}_l$, and $\mathbf{Z}_r$, respectively, the output of this multi-view generation is given by:

$$\hat{\mathbf{Z}} = \mathbf{Z}_b + \mathcal{C}(\mathbf{Z}_b, \mathbf{Z}_l, \mathbf{Z}_l) + \mathcal{C}(\mathbf{Z}_b, \mathbf{Z}_r, \mathbf{Z}_r), \tag{10}$$

where $\mathcal{C}(\cdot)$ represents the standard cross-attention operation, which accepts three input parameters: query, key, and value, respectively. This approach systematically integrates spatial information from various viewpoints, enabling the synthesis of images that exhibit visual consistency across distinct camera perspectives. Fig.7 offers a visual comparison of the model output with and without the application of view cross-attention. Upon integrating view cross-attention into PerLDiff, the procedure of the PerL-CM is detailed in Algo. 1.

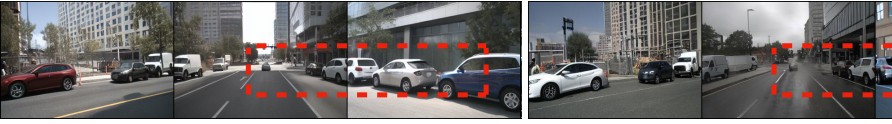

**w/ View Cross-attention**      **w/o View Cross-attention**

Figure 7: Comparative visualization of outputs with (left) and without (right) view cross-attention. Red markers highlight discontinuities in the images generated without view cross-attention.

## C.4 EFFECTIVENESS OF PERL-BASED CROSS ATTENTION (OBJECT)

To facilitate a better understanding of PerLDiff, we provide a detailed explanation of the PerL-based cross-attention (Object). As shown in Fig. 8, MagicDrive utilizes text cross-attention from Stable Diffusion to implicitly learn a unified feature that concatenates text, camera parameters, and bounding boxes in the token dimension. In contrast, PerLDiff employs the PerL masking map as a prior, allowing each object condition to precisely control the corresponding pixel features. This results in more accurate positioning and orientation of objects in the generated images. Additionally, we integrated the object mask into the token dimension corresponding to the bounding box. As shown in Tab. 8, the results indicate improvements in BEVFormer, with NDS (e.g., 29.77 vs. 28.79 for MagicDrive) and mAOE (e.g., 0.73 vs. 0.81 for MagicDrive) demonstrating the effectiveness of PerLDiff in enhancing the performance of MagicDrive. Note that MagicDrive utilizes a

single attention map for managing text, camera parameters, and boxes in the cross-attention process. Consequently, our ability to make improvements is constrained by the limited scope available for modifying the attention map within this architecture.

Table 8: Impact of integrating the PerL masking map (object) into MagicDrive. We present the 3D object detection results based on BEVFormer (Li et al., 2022), with outcomes showing superior performance emphasized in **bold**.

| Method | FID↓ | mAP↑ | NDS↑ | mAOE↓ | mAVE↓ | mATE↓ |
|---|---|---|---|---|---|---|
| MagicDrive | **15.92** | 15.21 | 28.79 | 0.81 | 0.57 | 0.95 |
| MagicDrive + Mask | 16.68 | **15.54** | **29.77** | **0.73** | **0.56** | **0.89** |

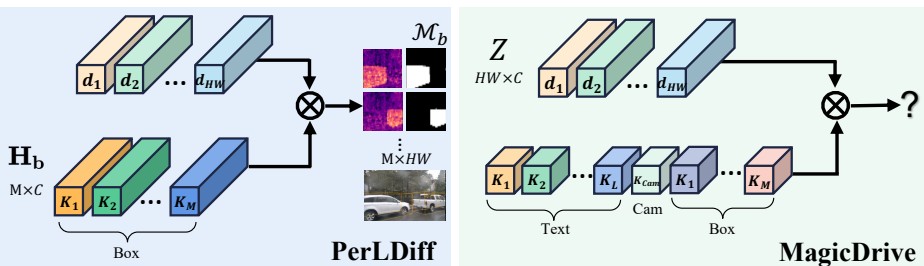

Figure 8: Overview of the PerL-based cross-attention (Object). MagicDrive employs text cross-attention to create a unified feature, while PerLDiff uses the PerL masking map to allow for precise control of pixel features for each object.

## C.5 EFFECTIVENESS OF MULTIPLICATION IN PERL-BASED CROSS ATTENTION

In Tab. 9, we conducted ablation experiments based on Equation 11. The results indicate that all metrics are lower than those of PerLDiff. We attribute this performance decline to the degradation of the cross-attention maps, which are highly sensitive to changes in the Perl masking map. As illustrated in Fig. 9, the cross attention maps learned by the Mu*Mask method excessively influence background regions, ultimately leading to a decline in image quality.

$$\mathcal{A}_s = softmax(\lambda_s \cdot \mathcal{M}_s * \mathbf{Z}\mathbf{H}_m^T/\sqrt{d}), \quad \mathcal{A}_b = softmax(\lambda_b \cdot \mathcal{M}_b * \mathbf{Z_s}\mathbf{H}_b^T/\sqrt{d}) \quad (11)$$

Table 9: Ablation experiments based on Equation 11. We report the 3D object detection results based on BEVFormer (Li et al., 2022), BEVFusion Liu et al. (2023) and BEV Segmentation results based on CVT. Outcomes demonstrating superior performance are highlighted in **bold**.

| Method | Detector | FID↓ | mAP↑ | NDS↑ | mAOE↓ | Road mIoU↑ | Vehicle mIoU↑ |
|---|---|---|---|---|---|---|---|
| Mu*Mask | BEVFusion | 15.00 | 13.95 | 22.68 | 0.84 | 56.42 | 26.82 |
| PerLDiff (Ours) | | **13.36** | **15.24** | **24.05** | **0.78** | **61.26** | **27.13** |
| Mu*Mask | BEVFormer | 15.00 | 23.01 | 33.62 | 0.78 | 56.42 | 26.82 |
| PerLDiff (Ours) | | **13.36** | **25.10** | **36.24** | **0.72** | **61.26** | **27.13** |

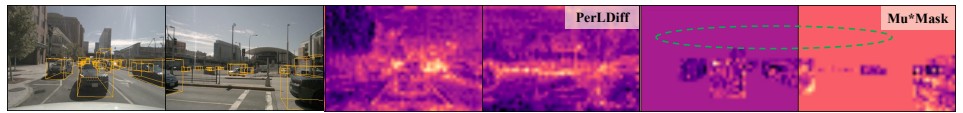

Figure 9: Cross-attention maps learned by the Mu*Mask method, with green circles indicating excessive influence on background regions, leading to a decline in image quality.

## C.6 EFFECTIVENESS OF PERLDIFF BASED ON CONTROLNET

In Tab. 10, we present an ablation study that replaces the architecture of PerLDiff with a ControlNet-based model trained only on view cross-attention in Stable Diffusion. As shown in Tab.10, the performance of the ControlNet-based model is inferior to that of PerLDiff. Furthermore, Fig. 10 illustrates that PerLDiff employs a network architecture similar to GLIGEN, allowing it to converge more quickly on smaller datasets, such as NuScenes, compared to the ControlNet architecture.

Table 10: Ablation study comparing PerLDiff with a ControlNet-based model. We present 3D object detection results based on BEVFormer, BEVFusion, and BEV segmentation results from CVT. Outcomes demonstrating superior performance are highlighted in **bold**.

| Method | Detector | FID↓ | mAP↑ | NDS↑ | mAOE↓ | Road mIoU↑ | Vehicle mIoU↑ |
|---|---|---|---|---|---|---|---|
| ControlNet + Mask | BEVFormer | 20.46 | 18.07 | 28.48 | 0.87 | 53.98 | 24.72 |
| PerLDiff (Ours) | | **13.36** | **25.10** | **36.24** | **0.72** | **61.26** | **27.13** |
| ControlNet + Mask | BEVFusion | 20.46 | 10.45 | 15.29 | 0.89 | 53.98 | 24.72 |
| PerLDiff (Ours) | | **13.36** | **15.24** | **24.05** | **0.78** | **61.26** | **27.13** |

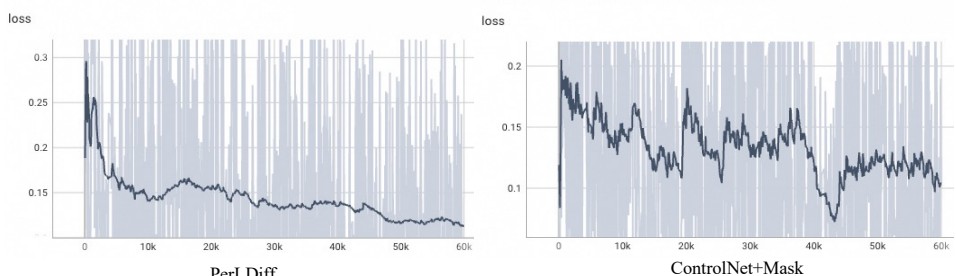

Figure 10: Training curves of PerLDiff and the ControlNet-based network, illustrating that PerLDiff converges more rapidly during training.

## D VISUALIZATION RESULTS

To further demonstrate the controllable generation capabilities of our PerLDiff, we present additional visual results. Fig. 11 offers extended examples illustrating the superiority of PerLDiff in scene controllability, while Fig. 12 highlights its effectiveness in controlling object orientation. Comparative visualizations, as illustrated in Fig. 13, reveal that BEVControl* produces chaotic and indistinct attention maps leading to suboptimal controllability, PerLDiff optimizes the response areas of the attention map, resulting in accurate object-level control. However, since PerLDiff incorporates prior constraints to ensure accuracy in object detection, this can adversely affect the details in the background of the images. As illustrated in Fig. 14, PerLDiff produces background details that do not align with those of real images.

Additionally, Fig. 15 displays scene alterations by PerLDiff to mimic different weather conditions or times of day, showcasing its versatility in changing scene descriptions. Moreover, we also demonstrate how PerLDiff can enhance the performance of the temporal-based model StreamPETR. It is worth noting that, based on our experimental results, the key for temporal-based detection models lies in accurately positioning and categorizing objects in each frame; detailed information about objects, such as color and brand, is not crucial. As illustrated in Fig. 16, when provided with continuous frame inputs, the generated images by PerLDiff ensure that the positions and categories of objects, along with the road map, are consistently aligned with the specified conditions between adjacent frames. Finally, Fig. 17 presents samples from KITTI *validation* set, illustrating the application's performance in real-world conditions.

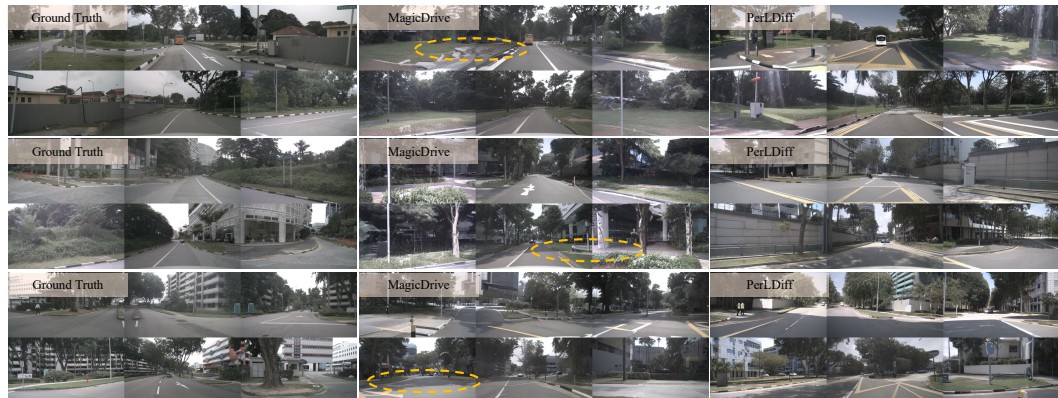

Figure 11: Qualitative comparison with MagicDrive. For **scene controllability**, PerLDiff demonstrates superior performance by generating images consistent with ground truth road information. Regions highlighted by yellow circles indicate areas where fail to align with ground truth conditions.

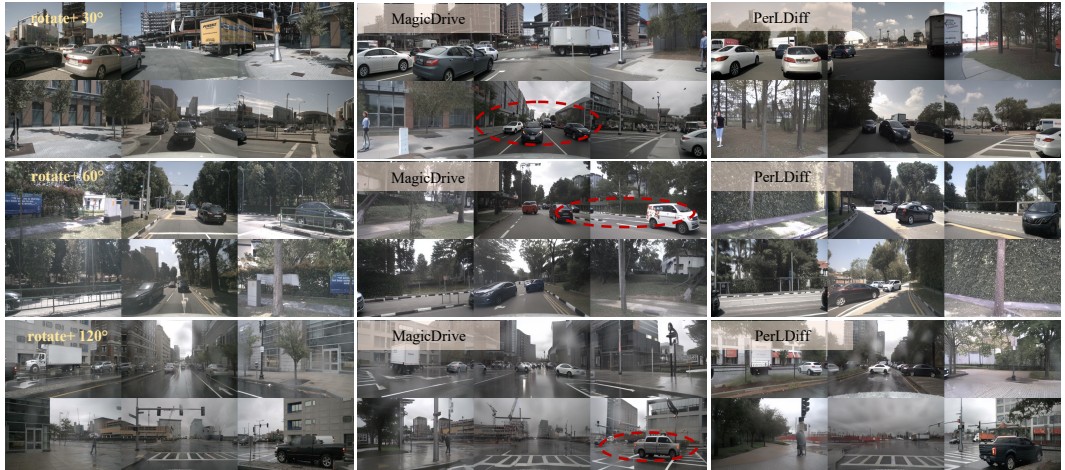

Figure 12: Qualitative comparison with MagicDrive. For **object controllability**, PerLDiff exhibits superior performance by generating objects at arbitrary angles. Regions highlighted by red circles denote scenarios where the generated images fail to achieve correct orientation.

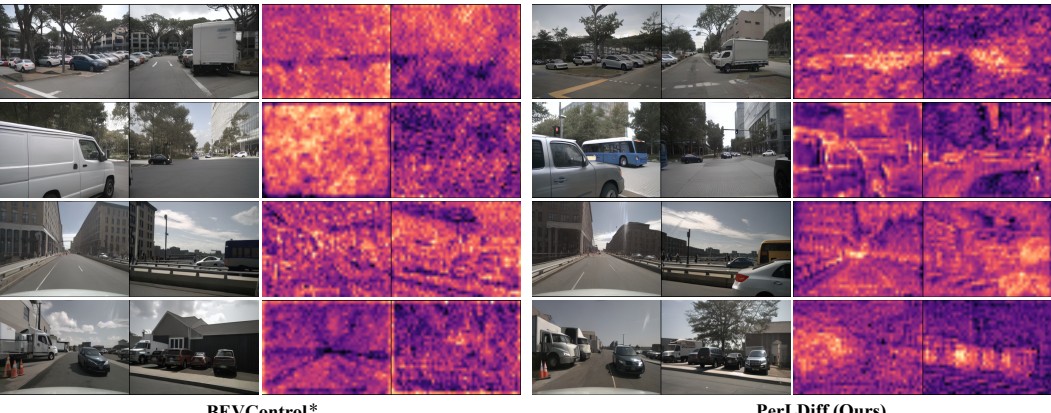

Figure 13: Visualization of cross-attention map results. From left to right, we present the generated images and corresponding cross-attention maps from our baseline BEVControl* and our PerLDiff.

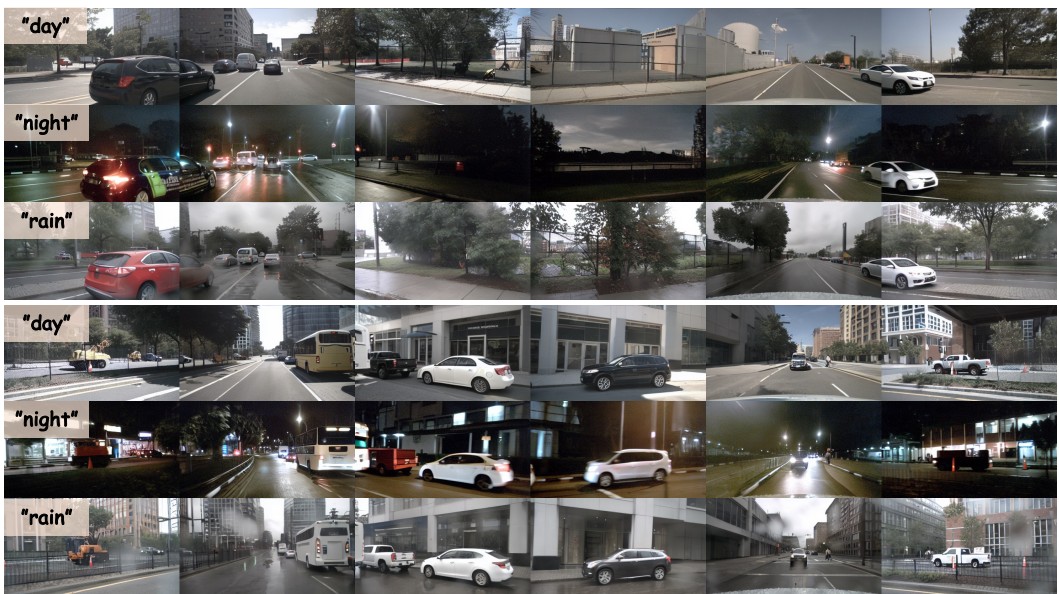

Figure 14: Qualitative visualizations of the generated images reveal discrepancies in background details. As indicated by the yellow circle, PerLDiff produces background elements that do not align with real images due to the incorporation of the PerL masking map.

Figure 15: Qualitative visualizations on NuScenes (Caesar et al., 2020): *day*, *night*, and *rain* scenarios synthesized by our PerLDiff, exhibiting adaptability to various lighting and weather conditions.

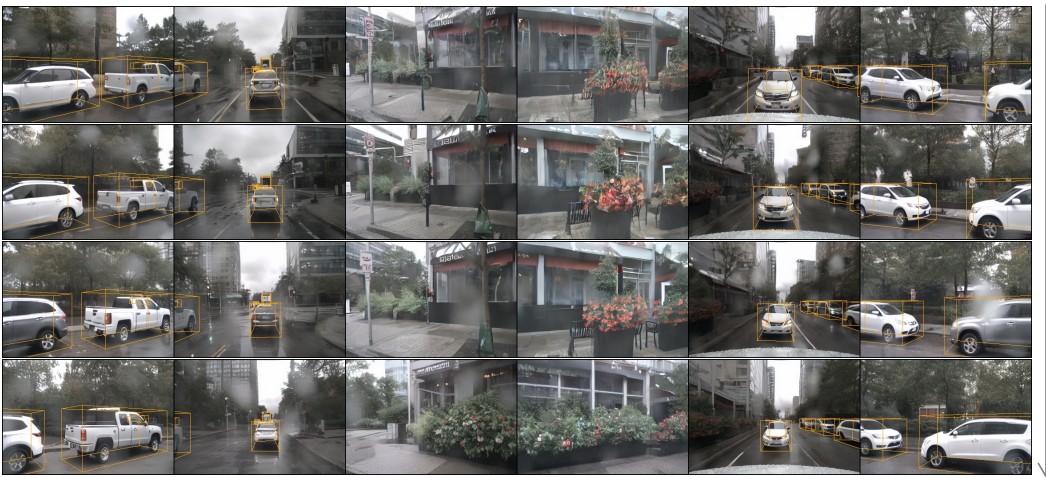

Figure 16: Qualitative visualizations from the NuScenes (Caesar et al., 2020). PerLDiff demonstrate consistent alignment of object positions and categories, along with the road map, when provided with continuous frame inputs, ensuring coherence between adjacent frames.

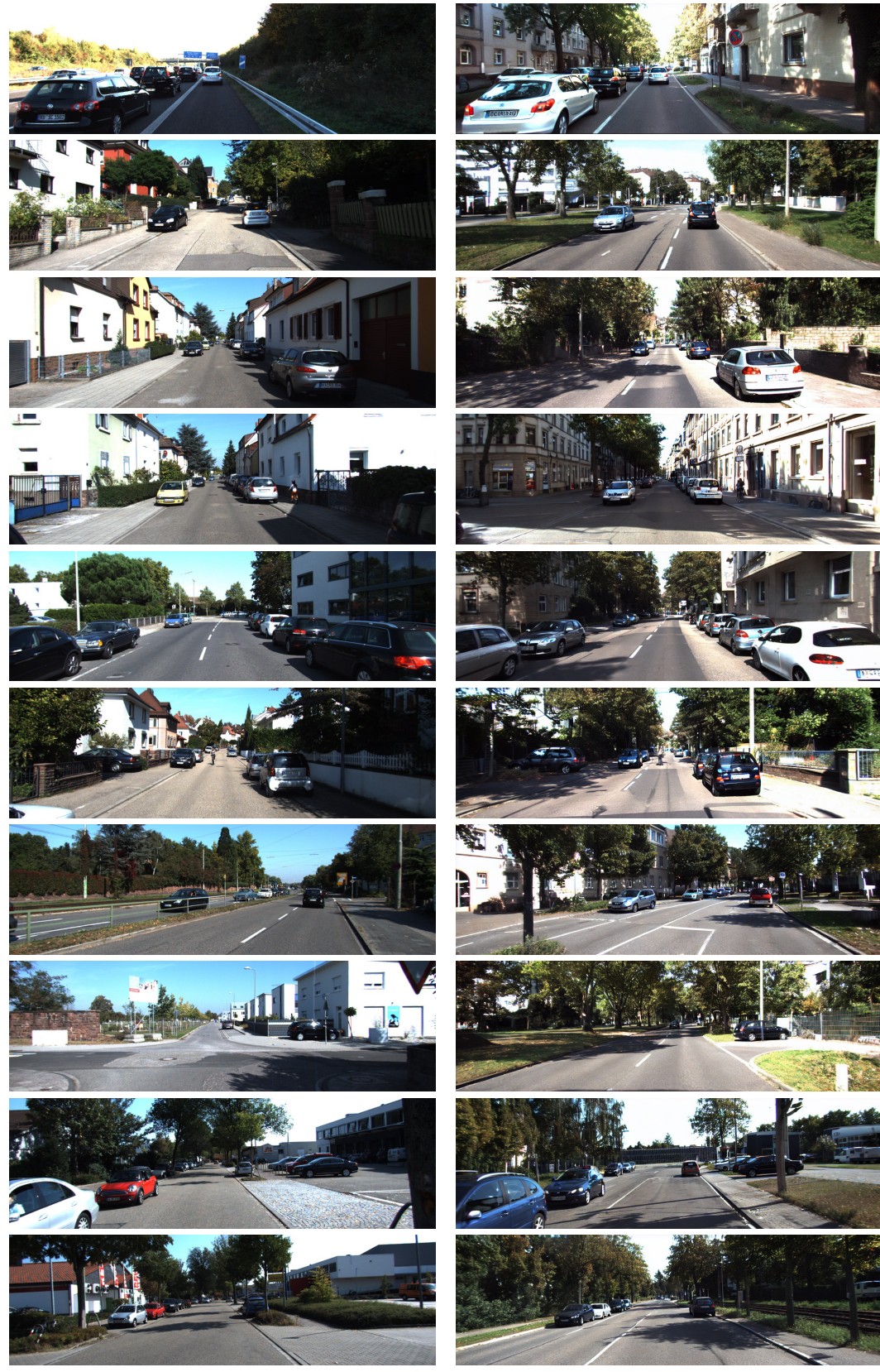

Figure 17: Visualization of street view images generated by our PerLDiff on KITTI (Geiger et al., 2012) *validation* dataset. We show the ground truth (left) and our PerLDiff (right).

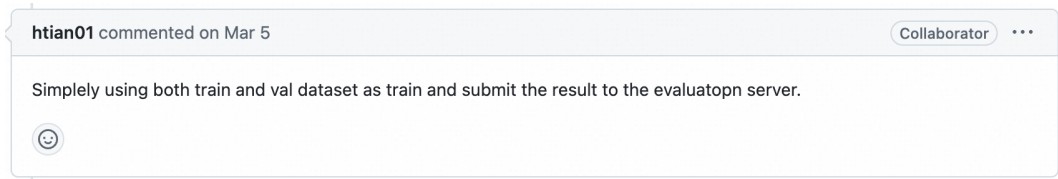

Figure 18: Screenshot of relevant GitHub issues illustrating that including the validation set in training is a common practice when dealing with small datasets.

