# OpenReview forum: "PerLDiff: Controllable Street View Synthesis Using Perspective-Layout Diffusion Model"
_ICLR.cc/2025/Conference — Submitted to ICLR 2025_

### Official Review · Reviewer_14qK · 2024-10-22

**Soundness:** 3
**Presentation:** 2
**Contribution:** 2
**Rating:** 3
**Confidence:** 4

**Summary:**

This paper introduces a controllable image generation framework specifically tailored for autonomous driving applications. It employs a modified attention module that enhances instance controllability by incorporating an instance binary mask, thereby improving the precision of instance manipulation within the generated images.

**Strengths:**

1. PerLDiff improves upon the controllability of previous work, enabling more effective control over vehicle pose and map segments.
2. This study validates its framework across diverse datasets and settings, encompassing the NuScenes dataset, the KITTI dataset, as well as 3D detection and map segmentation tasks.

**Weaknesses:**

1. The reason for selecting ConvNext as the feature extraction network requires further exploration, particularly given the training objective of image classification on the ImageNet dataset. This raises questions about the appropriateness of utilizing ConvNext in this context. Additionally, it is essential to conduct an ablation study comparing ConvNext with other image backbone architectures, such as the CLIP image encoder, to provide a comprehensive understanding of their relative performance.
2. The formulation of this work is not novel to me; rather, it represents a combination of several existing methodologies. The framework closely resembles that of Panacea and MagicDrive, with the key modification being the introduction of an attention mask.
3. Lack of quantitative comparison with Panacea [a], which similarly uses the perspective layout.

[a] Panacea: Panoramic and Controllable Video Generation for Autonomous Driving

**Questions:**

1. I suggest not using the word BEV annotations. Both 3D box annotations and map annotations exist within 3D space rather than BEV space.
2. Section 3.2 is hard to read. I suggest to include a dedicated diagram to effectively illustrate the PerL-based cross-attention mechanism. The bottom of Figure 2 need to be refine.
3. Why is the attention mask not multiplied into A_s in Equation 5? Adding a binary mask into the feature seems unconventional.
4. Given that using perspective-Layout, why not use controlnet framework for more straightforward interaction?
5. Can this method be applied to occupancy control?

---

> ### Author Response · Authors · 2024-11-22
> **Response to Reviewer 14qK by the Authors of PerLDiff (Part 1)**
>
> Dear Reviewer 14qK：
>
> Thank you for your review and comments. We provide the following discussions and explanations regarding your concerns.
>
> >**W1:** The reason for selecting ConvNext as the feature extraction network requires further exploration, particularly given the training objective of image classification on the ImageNet dataset. This raises questions about the appropriateness of utilizing ConvNext in this context. Additionally, it is essential to conduct an ablation study comparing ConvNext with other image backbone architectures, such as the CLIP image encoder, to provide a comprehensive understanding of their relative performance.
>
> **A1:**  We initially selected ConvNeXt due to its ability to effectively extract image features while maintaining a relatively low parameter count (27.82M). Furthermore, it has demonstrated efficacy in downstream segmentation tasks, particularly in extracting edge information from images [1].
>
> Moreover, we conducted an ablation study on the road map using the CLIP image encoder and the autoencoder from Stable Diffusion. The results indicate that the performance of the feature extraction network has negligible impact on PerLDiff ($\pm0.02$ NDS between CLIP encoder and ConvNext encoder).
>
> [1]. Liu, Zhuang et. al., A convnet for the 2020s. In CVPR2022.
>
> | Method                  | Detector  | FID↓      | mAP↑      | NDS↑      | mAOE↓    | Road mIoU↑ | Vehicle mIoU↑ |
> | ----------------------- | --------- | --------- | --------- | --------- | -------- | ---------- | ------------- |
> | CLIP encoder            |           | **12.77** | **25.41** | **36.26** | 0.73     | 60.49      | 27.06         |
> | VAE encoder             | BEVFormer | 18.55     | 24.88     | 35.28     | 0.75     | 53.24      | 26.94         |
> | **PerLDiff (ConvNext)** |           | 13.36     | 25.10     | 36.24     | **0.72** | **61.26**  | **27.13**     |
> | CLIP encoder            |           | **12.77** | **15.80** | **24.45** | 0.81     | 60.49      | 27.06         |
> | VAE encoder             | BEVFusion | 18.55     | 15.14     | 23.32     | **0.75** | 53.24      | 26.94         |
> | **PerLDiff (ConvNext)** |           | 13.36     | 15.24     | 24.05     | 0.78     | **61.26**  | **27.13**     |
>
>
>
> >**W2:** The formulation of this work is not novel to me; rather, it represents a combination of several existing methodologies. The framework closely resembles that of Panacea and MagicDrive, with the key modification being the introduction of an attention mask.
>
> **A2:** Our research is not merely a combination or replication of existing methods. While approaches like BEV Control, MagicDrive, and Panacea use BEV Layout as a condition and cross-attention for data generation, a fundamental question emerges: do these methods truly achieve conditional control? As Figure 1 in the paper shows, these methods sometimes fail in this regard. We propose that this failure arises because simple cross-attention does not guarantee a connection between control conditions and the generated images. As evidenced in Figure 3, simple cross-attention can produce disordered and blurry outputs, resulting in poor control performance.
>
> Based on this observation, we meticulously developed a PerL-based cross-attention mechanism that utilizes a PerL masking map—comprising road and box components—as geometric priors to facilitate high-precision, object-level image generation. This mechanism extracts conditional features from 3D annotations and explicitly renders them into perspective layout masking maps, serving as geometric priors.
>
> To validate the effectiveness of our approach, we conducted comprehensive experiments on the KITTI and NuScenes datasets, which demonstrated significant performance improvements over existing methods. Additionally, we performed data augmentation experiments to illustrate that our method can generate synthetic data for autonomous driving, offering a potential solution to address corner cases in this field.
>
>
>
> >**W3:**  Lack of quantitative comparison with Panacea [a], which similarly uses the perspective layout.
>
> **A3:** Since our PerLDiff  focuses on ensuring the controllability of objects in the generated images, we conducted comparisons with image generation models for validation (Penacea is a video generation method). Table 6 presents a comparison of various metrics, illustrating that our PerLDiff model can surpass the performance of Panacea even in the absence of temporal modeling.
>
> | Method              | Detector   | FID↓      | mAP↑      | NDS↑      | mAOE↓    | mATE↓ | mAVE↓    |
> | ------------------- | ---------- | --------- | --------- | --------- | -------- | ----- | -------- |
> | Oracle              |            |           | 47.05     | 56.24     | 0.37     | 0.61  | 0.27     |
> | Penacea             | StreamPETR | 16.96     | 22.50     | 36.10     | 0.73     | --    | 0.47     |
> | **PerLDiff (Ours)** |            | **15.67** | **35.09** | **44.19** | **0.64** | 0.75  | **0.45** |

---

> ### Author Response · Authors · 2024-11-22
> **Response to Reviewer 14qK by the Authors of PerLDiff (Part 2)**
>
> >**Q1:** I suggest not using the word BEV annotations. Both 3D box annotations and map annotations exist within 3D space rather than BEV space.
>
> **A4:** Thank you for your helpful comments. We have revised the term "BEV annotations" to "3D annotations" in our revision.
>
>
>
> >**Q2:** Section 3.2 is hard to read. I suggest to include a dedicated diagram to effectively illustrate the PerL-based cross-attention mechanism. The bottom of Figure 2 need to be refine.
>
> **A5:** To facilitate a better understanding of PerLDiff, we have provided a detailed explanation of PerL-based cross-attention (Object) in  **Figure 8 of Appendix C.4**. Compared to other methods, we utilize the Perl masking map as a prior, allowing each object condition to precisely control the corresponding pixel features. This results in more accurate positioning and orientation of objects in the generated images.
>
>
>
> >**Q3:** Why is the attention mask not multiplied into A_s in Equation 5? Adding a binary mask into the feature seems unconventional.
>
> **A6:** We conducted experiments based on Equation 1, as shown in Table 7 (Mu\*Mask). The results indicate that all metrics are lower than those of PerLDiff. We believe that the performance decline is due to the degradation of the cross-attention maps. Specifically, the cross-attention maps are highly sensitive to changes in the Perl masking map. As illustrated in **Figure 9 of Appendix C.5**, the attention maps learned by the Mu\*Mask method overly influence the background regions, ultimately leading to a decline in image quality.
> $$
> \mathcal{A}_s = \textit{softmax}(\lambda_s \cdot \mathcal{M}_s \times \mathbf{Z}\mathbf{H}_m^{T}/\sqrt{d}), \mathcal{A}_b = \textit{softmax}(\lambda_b \cdot \mathcal{M}_b \times \mathbf{Z_s}\mathbf{H}_b^{T}/\sqrt{d})
> $$
>
>
>
> | Method              | Detector  | FID↓      | mAP↑      | NDS↑      | mAOE↓    | Road mIoU↑ | Vehicle mIoU↑ |
> | ------------------- | --------- | --------- | --------- | --------- | -------- | ---------- | ------------- |
> | Mu*Mask             | BEVFormer | 15.00     | 23.01     | 33.62     | 0.78     | 56.42      | 26.82         |
> | **PerLDiff (Ours)** |           | **13.36** | **25.10** | **36.24** | **0.72** | **61.26**  | **27.13**     |
> | Mu*Mask             | BEVFusion | 15.00     | 13.95     | 22.68     | 0.84     | 56.42      | 26.82         |
> | **PerLDiff (Ours)** |           | **13.36** | **15.24** | **24.05** | **0.78** | **61.26**  | **27.13**     |
>
>
>
> >**Q4:** Given that using perspective-Layout, why not use controlnet framework for more straightforward interaction?
>
> **A7:** We adopted the ControlNet architecture, training view cross-attention within Stable Diffusion. As illustrated in Table 8 (ControlNet + Mask), the performance of models based on ControlNet is inferior to that of PerLDiff. Additionally, we found that PerLDiff employs a network architecture similar to GLIGEN, allowing for faster convergence on smaller datasets like NuScenes, compared to the ControlNet architecture, as shown in **Figure 10 of Appendix C.6**.
>
> | Method              | Detector  | FID↓      | mAP↑      | NDS↑      | mAOE↓    | Road mIoU↑ | Vehicle mIoU↑ |
> | ------------------- | --------- | --------- | --------- | --------- | -------- | ---------- | ------------- |
> | ControlNet + Mask   | BEVFormer | 20.46     | 18.07     | 28.48     | 0.87     | 53.98      | 24.72         |
> | **PerLDiff (Ours)** |           | **13.36** | **25.10** | **36.24** | **0.72** | **61.26**  | **27.13**     |
> | ControlNet + Mask   | BEVFusion | 20.46     | 10.45     | 15.29     | 0.89     | 53.98      | 24.72         |
> | **PerLDiff (Ours)** |           | **13.36** | **15.24** | **24.05** | **0.78** | **61.26**  | **27.13**     |
>
>
>
> >**Q5:** Can this method be applied to occupancy control?
>
> We hypothesize that your concern pertains to the efficacy of the data generated by PerLDiff for use in occupancy-based models. To address this, we tested the generated validation dataset using TPVFormer (CVPR 2023). As shown in Table 9, our results indicate a smaller discrepancy from the real data compared to MagicDrive and BEVControl, suggesting that the data generated by PerLDiff can also enhance occupancy-based segmentation networks.
>
> | Method      | Oracle | BEVControl | MagicDrive | PerLDiff (Ours) |
> | ----------- | ------ | ----------- | ----------- | --------------- |
> | Voxel mIoU↑ | 47.12  | 27.70       | 29.09       | **32.31**       |
> | Point mIoU↑ | 25.84  | 15.18       | 16.45       | **18.76**       |

---

> > ### Comment · Reviewer_14qK · 2024-11-25
> >
> > Thank you for the detailed reply. After reading the rebuttal, I have several follow-up questions.
> >
> > 1. With respect to A1, we all know that the variational autoencoder (VAE) features exhibit a higher degree of alignment with the latent diffusion process. However, this experiment appears to place it at a disadvantage when compared to its counterparts. This finding appears to be at odds with the conclusions drawn in the study ControlNeXt [a]. Can the author provide an analysis of this finding? Given that ConvNeXt is trained on an image classification objective, why does it achieve higher results in image quality and downstream application?
> >
> >
> > 2. Why do different tables exhibit different FID values?
> >
> >
> > 3. Regarding Q5, my question is whether PerLDiff can be used with other modalities to control image generation, for example, using depth, LiDAR, or occupancy to control image generation.
> >
> > [a] ControlNeXt: Powerful and Efficient Control for Image and Video Generation

---

> > > ### Author Response · Authors · 2024-11-25
> > >
> > > Dear Reviewer 14qK,
> > >
> > > Thank you for your feedback. Below is our response to your questions.
> > >
> > > >**Q1:**. With respect to A1, we all know that the variational autoencoder (VAE) features exhibit a higher degree of alignment with the latent diffusion process. However, this experiment appears to place it at a disadvantage when compared to its counterparts. This finding appears to be at odds with the conclusions drawn in the study ControlNeXt [a]. Can the author provide an analysis of this finding? Given that ConvNeXt is trained on an image classification objective, why does it achieve higher results in image quality and downstream application?
> > >
> > > **A1:** Although this module is not our innovation and we merely use a frozen feature extractor, we recognize the necessity of a thorough scientific discussion about the experiment's specifics, as requested by the reviewers. We concur with the reviewer that VAE features demonstrate better alignment with the latent diffusion process. However, these features exist in a high spatial dimension and must be compressed using a learnable MLP (H*W → 1) when using our PerL-based cross-attention mechanism, which inevitably disrupts feature alignment. A potential solution is to add the VAE-extracted features to the noisy image, as done in ControlNeXt and ControlNet. In this case, our PerL-based masking cannot be applied because no attention map is present. We have also conducted experiments incorporating this method, and the results are presented below.
> > >
> > > Our PerL-based masking method explicitly incorporates scene map information into the diffusion process, making it more efficient than traditional feature extraction and simple addition. Consequently, the "ConvNext encoder + PerL masking map (Scene&Object)" approach outperforms the "VAE encoder +  PerL masking map (Object) + Feature adding (Scene)" method.
> > >
> > >
> > > | Method                               | Detector  | FID↓      | mAP↑      | NDS↑      | mAOE↓    | Road mIoU↑ | Vehicle mIoU↑ |
> > > | ------------------------------------ | --------- | --------- | --------- | --------- | -------- | ---------- | ------------- |
> > > | CLIP encoder + PerL masking map (Scene&Object)      |           | **12.77** | **25.41** | **36.26** | 0.73     | 60.49      | 27.06         |
> > > | VAE encoder  + PerL masking map (Scene&Object)        | BEVFormer | 18.55     | 24.88     | 35.28     | 0.75     | 53.24      | 26.94         |
> > > | ConvNext encoder + PerL masking map (Scene&Object) |           | 13.36     | 25.10     | 36.24     | **0.72** | **61.26**  | **27.13**     |
> > > | VAE encoder +  PerL masking map (Object) + Feature adding (Scene) |           | 13.21     | 24.93     | 35.76     | 0.74     | 60.87      | 27.07         |
> > >
> > > >**Q2:** Why do different tables exhibit different FID values?
> > >
> > > **A2:** In Table 6, StreamPETR utilizes a resolution of 256$\times$704, while Penacea operates at a resolution of 256$\times$512. Therefore, we conducted our comparison at the higher resolution used by StreamPETR, which resulted in different FID scores. Furthermore, as outlined in **Table 6 of Appendix C.1** of this paper, we also compared our results with those of MagicDrive at the same resolution.
> > >
> > > >**Q3:** Regarding Q5, my question is whether PerLDiff can be used with other modalities to control image generation, for example, using depth, LiDAR, or occupancy to control image generation.
> > >
> > > **A3:** Our motivation for developing PerLDiff stems from the necessity to address the persistent challenge of data scarcity in autonomous driving. Consequently, obtaining and modifying the input depth, LiDAR, or occupancy data prior to generation is impractical for synthetic data generation. For further details, please refer to our response to Review sheB in W1, where we discuss the methodologies and motivations behind our work.

---

> > > > ### Author Response · Authors · 2024-11-27
> > > > **Request for Review 14qK Feedback**
> > > >
> > > > Dear Reviewer 14qK,
> > > >
> > > > We sincerely thank you for your insightful review and trust that our response effectively addresses your previous concerns regarding this paper.
> > > >
> > > > In response to your specific issues: in Appendix (Sec. C.4), we have supplemented our paper with additional ablation experiments and diagram to clarify our motivation. In Appendix (Sec. C.5 and C.6), we conducted experiments utilizing the settings "multiplied into
> > > > A_s in Equation 5" and "utilizing the ControlNet framework" to validate the effectiveness of our method. In Table 5, we included further ablation studies to demonstrate the impact of different choices of road map encoders; in Table 6, we conducted comprehensive comparisons with Penacea to highlight the advantages of our proposed algorithm. Lastly, we clarified our motivation for addressing the persistent challenge of data scarcity, emphasizing the impracticality of obtaining and modifying input depth, LiDAR, or occupancy data prior to generation.
> > > >
> > > > As the discussion period draws to a close, we kindly invite you to share any additional comments or concerns about our work; we would be more than happy to address them. In the meantime, we hope you might consider revisiting your evaluation and potentially increasing your rating.
> > > >
> > > > Thank you for your thoughtful feedback and consideration; we truly appreciate it!
> > > >
> > > > Best regards,
> > > >
> > > > Paper 52 Authors

---

> ### Author Response · Authors · 2024-12-02
> **Request for Review 14qK Feedback Again**
>
> Dear Reviewer 14qK,
>
> We would like to gently remind you that **today is the last day for reviewers to post messages to the authors**, and we have not yet received any additional feedback from you since November 25. If our responses and revisions can help address your concerns, we kindly request that you consider updating your rating.
>
> Your insights are invaluable to us, and we would greatly appreciate it if you could share your thoughts before the end of the day.
>
> Thank you for your time and consideration. We look forward to your response.
>
> Best regards,
>
> The Authors of Paper 52

---

### Official Review · Reviewer_gVD5 · 2024-11-03

**Soundness:** 3
**Presentation:** 3
**Contribution:** 3
**Rating:** 6
**Confidence:** 4

**Summary:**

This paper presents a novel method for controllable street view generation utilizing a perspective layout diffusion model. The control factors are formulated as bounding boxes, lane structures, and scene descriptions. Subsequently, a cross-view framework, grounded in stable diffusion, is crafted to synthesize the desired multi-view images. Experimental results across various settings (i.e. generation quality, 3D object detection, and lane segmentation performance)—demonstrate the efficacy of the proposed approach.

**Strengths:**

+  solution for the data hungry autonomous driving. Meanwhile, the authors also conduct experiments to demonstrate the generated data can improve perception models' performance.

+ The writing is clear and the paper is easy to follow.

+ The authors conduct multiple experiments under different settings to demonstrate the proposed method can outperform the previous view synthesis methods.

**Weaknesses:**

+ The first question is whether the proposed method can ensure temporal consistency for the generated images. The paper doesn't discuss this problem and displays the image in different timestamps. However, the detection and segmentation models used, particularly StreamPETR, require temporal inputs. If the synthesized images do not achieve good temporal consistency, how can they improve the performance of temporal-based methods?


+ In Section 3, it appears that all input data comes from 2D space (with boxes and roadmaps projected onto images and a general scene description). Another question is whether the proposed method can effectively disentangle the camera's intrinsic and extrinsic parameters compared to BEVGen and BEVControl.


+ Cross-view attention: Could the authors explain the differences between view cross-attention and text cross-attention in PerlDiff compared to BEVControl and MagicDrive?


+ Object rotation controllability: Could the authors clarify how the proposed method achieves object rotation controllability? In Section 3, it seems that the box information is represented as eight corners in the image space, where the rotation information is not explicitly presented.

**Questions:**

+ Could the authors provide the details of the experimental setting on the nuScenes dataset? One point of confusion is that,  compared to the training set, the validation set is not very large. Why does adding it to the training data lead to such a significant improvement in model performance?

+ What is the meaning of * in Table 3?

---

> ### Author Response · Authors · 2024-11-22
> **Response to Reviewer gVD5 by the Authors of PerLDiff (Part 1)**
>
> Dear Reviewer gVD5:
>
> Thank you for your acknowledgment and constructive comments. We provide discussions and explanations about your concerns as follows.
>
> >**W1:** The first question is whether the proposed method can ensure temporal consistency for the generated images. The paper doesn't discuss this problem and displays the image in different timestamps. However, the detection and segmentation models used, particularly StreamPETR, require temporal inputs. If the synthesized images do not achieve good temporal consistency, how can they improve the performance of temporal-based methods?
>
> **A1:** We acknowledge that PerLDiff currently does not incorporate any temporal blocks for video generation. It is worth noting that, based on our experimental results, the key for temporal-based detection models is the positions and categories of objects in each frame; detailed information about objects, such as color and brand, is not crucial. As illustrated in **Figure 16 of Appendix D**, when provided with continuous frame inputs, the generated images ensure that the positions and categories of objects, as well as the road map, are accurately consistent with the specified conditions between adjacent frames.
>
>
>
> >**W2:** In Section 3, it appears that all input data comes from 2D space (with boxes and roadmaps projected onto images and a general scene description). Another question is whether the proposed method can effectively disentangle the camera's intrinsic and extrinsic parameters compared to BEVGen and BEVControl.
>
> **A2:** We did not fully understand your comments and would appreciate your feedback on our response below. All annotations are projected from 3D space using both intrinsic and extrinsic parameters. BEVGen, BEVControl, and our PerLDiff all incorporate these parameters to process object coordinates due to the task setup.
>
>
>
> >**W3:** Cross-view attention: Could the authors explain the differences between view cross-attention and text cross-attention in PerlDiff compared to BEVControl and MagicDrive?
>
> **A3:** View cross-attention and text cross-attention are common settings. For view cross-attention, as described in Lines 301-304, MagicDrive, BEVControl, DrivingDiffusion, Panacea, and PerLDiff employ the same mechanism. The key idea is to obtain features from adjacent views to ensure consistency. For text cross-attention, BEVControl and MagicDrive freeze the weights of Stable Diffusion, while PerLDiff trains the text cross-attention block to enhance scene control.
>
>
>
> >**W4:** Object rotation controllability: Could the authors clarify how the proposed method achieves object rotation controllability? In Section 3, it seems that the box information is represented as eight corners in the image space, where the rotation information is not explicitly presented.
>
> **A4:** PerLDiff controls the rotation of objects through the input coordinates, as illustrated in the figure below. As you pointed out, we utilize the eight corner points arranged in **sequential order, with the first four points representing the front of the vehicle**. This effectively encapsulates the angle information of the objects. Additionally, we use the object's Perl masking map as a geometric prior to ensure that the generated object angles align correctly.
>
> ```markdown
>                                                                         up z
>                                                       front x           ^
>                                                            /            |
>                                                           /             |
>                                          	         <1>-------------<2>
>                                                         /|            / |
>                                                        / |           /  |
>                                        		     <5>------------<6>   <3>
>                                                       |  /          |  /
>                                                       | / origin    | /
>                                       left y<-------- <8> ----------<7>
>
> ```

---

> ### Author Response · Authors · 2024-11-22
> **Response to Reviewer gVD5 by the Authors of PerLDiff (Part 2)**
>
> >**Q1:**  Could the authors provide the details of the experimental setting on the nuScenes dataset? One point of confusion is that, compared to the training set, the validation set is not very large. Why does adding it to the training data lead to such a significant improvement in model performance?
>
> **A5:** PerLDiff is trained using the training set from nuScenes, which consists of a total of 28,130 samples across six views. We set the batch size to 16 (2 per GPU) and the learning rate to 5e-5, training for a total of 60,000 iterations. In Table 3, for the training of all BEVFormer models, we adopted the official configuration with a batch size of 1 and 24 epochs, only adjusting the resolution to 256x384. For training all StreamPETR models, we utilized the official configuration, employing a batch size of 2 and running for 24 epochs. Additional training details are provided in Appendix B.
>
> The improvement observed in the test set performance after incorporating the validation set into the training process (comprising 28,130 training frames and 6,019 validation frames) can be attributed to the relatively limited size of the original training dataset. Including the validation set, which increases the data by 21.4\%, is a common practice when dealing with small datasets. For further reference, a screenshot of relevant GitHub issues is provided in **Figure 18 of Appendix D**.
>
>
>
> >**Q2:**  What is the meaning of * in Table 3?
>
> **A6:** In Table 3, it should be “Syn. \textit{val}\*” represents the synthetic \textit{validation} set generated by BEVControl. We have corrected these typos in the revision.

---

> ### Author Response · Authors · 2024-11-27
> **Request for Review gVD5 Feedback**
>
> Dear Reviewer gVD5,
>
> We sincerely thank you for your insightful review and hope that our response effectively addresses your previous concerns regarding this paper.
>
> In response to the issue you raised: in Figure 16 of Appendix D, we have supplemented the visual analysis and provided an explanation of the key elements for temporal-based detection models. Additionally, in our comments, we included a detailed explanation of your other questions regarding our work, such as "Object rotation controllability" and the "details of the experimental setting."
>
> As the discussion period comes to a close, we kindly invite you to share any additional comments or concerns about our work. We would be more than happy to address them. In the meantime, we hope you might consider revisiting your evaluation and potentially increasing your rating.
>
> Thank you for your thoughtful feedback and consideration! We truly appreciate it!
>
> Best regards,
>
> The Authors of Paper 52

---

> > ### Author Response · Authors · 2024-12-02
> > **Request for gVD5 Feedback Again**
> >
> > Dear Reviewer gVD5,
> >
> > We would like to gently remind you that **today is the last day for reviewers to post messages to the authors**, and we have not yet received any additional feedback from you since November 22. If our responses and revisions can help address your concerns, we kindly request that you consider updating your rating.
> >
> > Your insights are invaluable to us, and we would greatly appreciate it if you could share your thoughts before the end of the day.
> >
> > Thank you for your time and consideration. We look forward to your response.
> >
> > Best regards,
> >
> > The Authors of Paper 52

---

### Official Review · Reviewer_sheB · 2024-11-04

**Soundness:** 3
**Presentation:** 4
**Contribution:** 3
**Rating:** 6
**Confidence:** 5

**Summary:**

This paper introduces PerLDiff, a method aimed at enhancing control in street view image generation, which is crucial for efficient data annotation in autonomous driving applications. PerLDiff incorporates layout-based masking maps as geometric priors and introduces a PerL-based Cross-Attention Mechanism within the Control Module (PerL-CM). This mechanism facilitates precise alignment of objects and scene details by integrating scene-wide and object-specific information derived from BEV annotations. Empirical evaluations on NuScenes and KITTI datasets suggest that PerLDiff achieves improved control and realism compared to baseline models such as BEVControl and MagicDrive.

**Strengths:**

* The overall paper writing is clear and is easy to follow.
* The introduced PerL-based Cross-attention sounds reasonable and performs well on some detailed experiments.
* PerLDiff demonstrates better object position controllability in scene generation compared to methods like BEVControl and MagicDrive.

**Weaknesses:**

* The authors use a mask-based representation in the PerL-based Cross-Attention mechanism. However, this type of representation lacks sufficient 3D priors, which may contribute to the orientation generation issues for vehicles mentioned in the limitations. Did the authors consider using the depth information of multiple vehicles with ControlNet to enhance object controllability? Specifically, it would be insightful to explore the benefits of Depth + ControlNet vs Mask + PerL-based Cross-Attention in improving object controllability.
* Has the PerL-based Cross-Attention been tested on other baselines beyond BEVControl to examine its potential improvements in controllability and perception tasks? For instance, would applying it to a baseline like MagicDrive yield similar gains?
* Is there any ablation study on the ConvNext for encoding the road map? How about other choices for the road map encoding? And why the ConvNext is frozen?

**Questions:**

* The term "train + Syn. val*" in Table 3 is not clearly defined.
* In Table 1, why the FID metric of PerLDiff is worse than the baseline BEVControl? Moreover, why the performance with BEVFormer is better than the one with multi-modal BEVFusion?
* In Table 2, why PerLDiff is significantly better than the BEVControl? I want to know the true reason behind the method perspective.

---

> ### Author Response · Authors · 2024-11-22
> **Response to Reviewer sheB by the Authors of PerLDiff (Part 1)**
>
> Dear Reviewer sheB:
>
> Thank you for your acknowledgment of PerLDiff’s idea and constructive comments. We provide the following discussions and explanations regarding your concerns.
>
> >**W1:** The authors use a mask-based representation in the PerL-based Cross-Attention mechanism. However, this type of representation lacks sufficient 3D priors, which may contribute to the orientation generation issues for vehicles mentioned in the limitations. Did the authors consider using the depth information of multiple vehicles with ControlNet to enhance object controllability? Specifically, it would be insightful to explore the benefits of Depth + ControlNet vs Mask + PerL-based Cross-Attention in improving object controllability.
>
> **A1:** Firstly, in addition to the PerL masking map, we also utilize the **sequential eight corner** coordinates of each object's bounding box as a controlling condition, representing **sparse 3D prior knowledge**. Secondly, in accordance with your suggestion, we implemented depth maps as controlling conditions. As demonstrated in Table 1 (Depth + ControlNet), their effectiveness is inferior to that of PerLDiff. Finally, one motivation for PerLDiff is to mitigate the issue of data scarcity. It is often i**mpractical to obtain and manipulate the depth maps of multiple vehicles** from images for synthetic data generation.
>
> | Method              | Detector  | FID↓      | mAP↑      | NDS↑      | mAOE↓    | Road mIoU↑ | Vehicle mIoU↑ |
> | ------------------- | --------- | --------- | --------- | --------- | -------- | ---------- | ------------- |
> | Depth + ControlNet  | BEVFormer | 24.48     | 15.36     | 32.36     | **0.66** | 59.27      | 24.46         |
> | **PerLDiff (Ours)** |           | **13.36** | **25.10** | **36.24** | 0.72     | **61.26**  | **27.13**     |
> | Depth + ControlNet  | BEVFusion | 24.48     | 6.62      | 16.64     | 0.82     | 59.27      | 24.46         |
> | **PerLDiff (Ours)** |           | **13.36** | **15.24** | **24.05** | **0.78** | **61.26**  | **27.13**     |
>
>
>
>
> >**W2:** Has the PerL-based Cross-Attention been tested on other baselines beyond BEVControl to examine its potential improvements in controllability and perception tasks? For instance, would applying it to a baseline like MagicDrive yield similar gains?
>
> **A2:** As illustrated in **Figure 8 of Appendix C.4**, MagicDrive utilizes text-based cross-attention from Stable Diffusion to implicitly learn a unified feature that concatenates text, camera parameters, and bounding boxes in the token dimension. Based on your suggestion, we integrated the object mask into the token dimension corresponding to the bounding box. As shown in Table 2 (MagicDrive + Mask), results indicate improvements on BEVFormer, with NDS (e.g., 29.77 vs. 28.79 for MagicDrive) and mAOE (e.g., 0.73 vs. 0.81 for MagicDrive) demonstrating the effectiveness of PerLDiff in enhancing the performance of MagicDrive. Note that MagicDrive utilizes a single attention map for managing text, camera parameters, and boxes in the  cross-attention process. Consequently, our ability to make improvements is constrained by the limited scope available for modifying the attention map within this architecture.
>
> | Method        |     FID↓      | mAP↑      | NDS↑      | mAOE↓    | mAVE↓    | mATE↓    |
> | ----------------- | -------- | --------- | --------- | --------- | -------- | -------- |
> | MagicDrive       | **15.92** | 15.21     | 28.79     | 0.81     | 0.57     | 0.95     |
> | MagicDrive + Mask |  16.68     | **15.54** | **29.77** | **0.73** | **0.56** | **0.89** |

---

> ### Author Response · Authors · 2024-11-22
> **Response to Reviewer sheB by the Authors of PerLDiff (Part 2)**
>
> >**W3:** Is there any ablation study on the ConvNext for encoding the road map? How about other choices for the road map encoding? And why the ConvNext is frozen?
>
> **A3:** Following your suggestion, we conducted an ablation study on the road map using the CLIP image encoder and the autoencoder from Stable Diffusion. The results reveal that the performance of the feature extraction network has negligible impact on PerLDiff ($\pm0.02$ NDS between CLIP encoder and ConvNext encoder). The reason for freezing ConvNext is twofold: it reduces the number of parameters to be trained and, since models like ConvNext already possess strong feature extraction capabilities, additional training is unnecessary.
>
> | Method                  | Detector  | FID↓      | mAP↑      | NDS↑      | mAOE↓    | Road mIoU↑ | Vehicle mIoU↑ |
> | ----------------------- | --------- | --------- | --------- | --------- | -------- | ---------- | ------------- |
> | CLIP encoder            |           | **12.77** | **25.41** | **36.26** | 0.73     | 60.49      | 27.06         |
> | VAE encoder             | BEVFormer | 18.55     | 24.88     | 35.28     | 0.75     | 53.24      | 26.94         |
> | **PerLDiff (ConvNext)** |           | 13.36     | 25.10     | 36.24     | **0.72** | **61.26**  | **27.13**     |
> | CLIP encoder            |           | **12.77** | **15.80** | **24.45** | 0.81     | 60.49      | 27.06         |
> | VAE encoder             | BEVFusion | 18.55     | 15.14     | 23.32     | **0.75** | 53.24      | 26.94         |
> | **PerLDiff (ConvNext)** |           | 13.36     | 15.24     | 24.05     | 0.78     | **61.26**  | **27.13**     |
>
>
>
> >**Q1:** The term "train + Syn. val*" in Table 3 is not clearly defined.
>
> **A4:** In Table 3, it should be “Syn. \textit{val}\*” represents the synthetic \textit{validation} set generated by BEVControl. We have corrected these typos in the revision.
>
>
>
> >**Q2:** In Table 1, why the FID metric of PerLDiff is worse than the baseline BEVControl? Moreover, why the performance with BEVFormer is better than the one with multi-modal BEVFusion?
>
> **A5:** Regarding the FID metric, our results are comparable to those of BEVControl. While PerLDiff incorporates prior constraints to ensure accuracy in object detection, this may adversely affect the details in the background of the images. As illustrated in **Figure 14 of Appendix D**, PerLDiff produces background details that do not align with those of real images.
>
> As for the performance of BEVFusion, we apologize for the confusion. We utilized the BEVFusion (Camera Only) model, similar to MagicDrive, which indeed shows performance metrics weaker than those of BEVFormer. Additionally, we present the results of BEVFusion (Camera + LiDAR) in Table 4, where the performance metrics are significantly higher than those of BEVFormer.
>
> | **Detector**                | **mAP**↑ | **NDS**↑ | **mAOE**↓ | **mATE**↓ |
> |-----------------------------|----------|----------|----------|----------|
> | BEVFormer                  | 25.10    | 36.24    | 0.72      | 0.84      |
> | BEVFusion (Camera Only)    | 15.24    | 24.05    | 0.78      | 0.98      |
> | BEVFusion (Camera + LiDAR) | 66.73 | 70.49 | 0.31 | 0.29 |
>
>
>
>
> >**Q3:** In Table 2, why PerLDiff is significantly better than the BEVControl? I want to know the true reason behind the method perspective.
>
> **A6:** Table 2 presents the comparative performance results of monocular 3D object detection using various synthetic KITTI images. There are two main reasons for the observed differences. **First**, the limited size of the KITTI dataset, which contains just 3,712 training images, impedes the learning process of traditional methods that do not utilize PerL masking map. **Second**, monocular 3D object detection is highly sensitive to accurate depth prediction. Depth is derived from the 2D projected size and the estimated 3D size through perspective projection. Our method produces more precise object sizing, thereby enhancing detection performance.

---

> ### Author Response · Authors · 2024-11-27
> **Request for Review sheB Feedback**
>
> Dear Reviewer sheB,
>
> We sincerely thank you for your insightful review and hope that our response effectively addresses your previous concerns regarding this paper.
>
> Regarding the issue you are concerned about: we have included additional experiments regarding "Depth + ControlNet" in Table 1. In Table 3, we provided further ablation studies to demonstrate the impact of different choices of road map encoders. In Table 4, we explain the reasons for the performance gap with BEVFormer and BEVFusion. In Appendix (Sec. C.4), we supplemented additional experiments on MagicDrive and diagram to clarify our motivation. Additionally, in Figure 14 of Appendix (Sec. D), we provided visual examples to explain the FID metric.
>
> As the discussion period draws to a close, we kindly invite you to share any additional comments or concerns about our work. We would be more than happy to address them. In the meantime, we hope you might consider revisiting your evaluation and potentially increasing your rating.
>
> Thank you for your thoughtful feedback and consideration! We truly appreciate it!
>
> Best regards,
>
> The Authors of Paper 52

---

> > ### Comment · Reviewer_sheB · 2024-11-29
> >
> > Thanks for the response from the authors. After reading the rebuttal, I have several questions.
> >
> > For W1, I feel it may introduce some confusion to the authors. Actually, I want to see if introducing depth information of objects with ControlNet can bring performance improvements for PerDiff instead of adding the experiment of Depth + ControlNet.
> >
> > For W2, I notice that the MagicDrive + Mask brings very marginal improvement to MagicDrive, which makes me doubt the effectiveness of PerL-CM on stronger baseline approaches. The authors claim that the improvement is constrained by the single attention map of MagicDrive. However, it also proves its lower generalization of PerDiff. Moreover, how about the setting of baseline MagicDrive here? I found the performance is located between 224×400 and 272×736 resolutions in original MagicDrive paper.
> >
> > For W3, the ablation study for ConvNext has several questions. Overall, it shows CLIP encoder outperforms ConvNext? Why the performance of BEVFormer and BEVFusion on mIoU metrics are the same?
> >
> > I would like the authors to further clarify those questions.

---

> > > ### Author Response · Authors · 2024-12-02
> > > **Response to Reviewer sheB’s Feedback (Part-1)**
> > >
> > > Dear Reviewer sheB,
> > >
> > > Thank you for your valueable feedback. Below is our response to your questions.
> > >
> > > > **Q1:** For W1, I feel it may introduce some confusion to the authors. Actually, I want to see if introducing depth information of objects with ControlNet can bring performance improvements for PerDiff instead of adding the experiment of Depth + ControlNet.
> > >
> > > **A1:** Based on your suggestion, we utilized Marigold[1] (CVPR 2024 Oral) to generate the depth map and then trained ControlNet to incorporate depth map information into the frozen PerLDiff model. We trained ControlNet using the same settings as those employed for training PerLDiff, with a batch size of 16 (2 per GPU) and 60,000 iterations. As shown in Table, we observed slight improvements in the metrics across BEVFormer, BEVFusion, and CVT, particularly in mAOE of BEVFormer (e.g., 0.72 vs. 0.61 for PerLDiff + Depth). This indicates that the inclusion of depth maps can partially enhance the performance of PerLDiff.
> > >
> > > [1]Marigold: Ke, Bingxin, et al. "Repurposing diffusion-based image generators for monocular depth estimation." Proceedings of the IEEE/CVF Conference on Computer Vision and Pattern Recognition. 2024.
> > >
> > > | Method               | Detector  | mAP↑      | NDS↑      | mAOE↓    | mATE↓    |
> > > | -------------------- | --------- | --------- | --------- | -------- | -------- |
> > > | PerLDiff             | BEVFormer | 25.10     | 36.24     | 0.72     | 0.84     |
> > > | **PerLDiff + Depth** |           | **25.27** | **38.13** | **0.61** | **0.80** |
> > > | PerLDiff             | BEVFusion | 15.24     | 24.05     | 0.78     | 0.88    |
> > > | **PerLDiff + Depth** |           |     **15.38**      |   **24.54**       |    **0.74**      |     **0.82**     |
> > >
> > > | Method               | Segmentation | Road mIoU↑ | Vehicle mIoU↑ |
> > > | -------------------- | -------- | ---------- | ------------- |
> > > | PerLDiff             | CVT      | 61.26      | 27.13         |
> > > | **PerLDiff + Depth** |          | **62.77**  | **27.18**     |
> > >
> > >
> > >
> > > > **Q2:** For W2, I notice that the MagicDrive + Mask brings very marginal improvement to MagicDrive, which makes me doubt the effectiveness of PerL-CM on stronger baseline approaches. The authors claim that the improvement is constrained by the single attention map of MagicDrive. However, it also proves its lower generalization of PerDiff. Moreover, how about the setting of baseline MagicDrive here? I found the performance is located between 224×400 and 272×736 resolutions in original MagicDrive paper.
> > >
> > > **A2: First, 1)**: As described above, MagicDrive utilizes text, camera parameters, and bounding box coordinates as unified conditional features. After linear projection in cross-attention, the various conditional features in the obtained keys (K) and values (V) do not function independently to control image generation; thus, the naive attempt to merely add a perl masking map (box) to the bounding box dimensions will not yield significant improvements due to this unified feature dependency. **2):** Our goal is not to construct a plug-and-play module for enhancing all methods; rather, we focus on developing a refined and effective approach to enhance object and scene controllability in the current task, ultimately increasing its practical applicability.
> > >
> > >
> > >
> > > **Second**, we utilized the setup from MagicDrive available on GitHub. Since only the training and testing code for a resolution of 224x400 was published, we conducted our experiments solely at this resolution while keeping all other settings unchanged.

---

> > > > ### Author Response · Authors · 2024-12-02
> > > > **Response to Reviewer sheB’s Feedback (Part-2)**
> > > >
> > > > > **Q3:** For W3, the ablation study for ConvNext has several questions. Overall, it shows CLIP encoder outperforms ConvNext? Why the performance of BEVFormer and BEVFusion on mIoU metrics are the same?
> > > >
> > > > **A3:** **First**, as illustrated in Table, the comparison between the CLIP image encoder and ConvNext reveals a negligible impact on the PerLDiff module. While the CLIP image encoder slightly outperforms ConvNext in detection metrics, ConvNext shows a marginal advantage over the CLIP image encoder in segmentation tasks. Furthermore, it is important to note that this module is not our innovation; we utilize it solely as a frozen feature extractor.
> > > >
> > > > **Second**, ConvNext has a parameter count that is only approximately 9.15\% of that of the CLIP image encoder (e.g., **27.82M vs. 303.97M**). Thus, to achieve a balance between performance and GPU utilization, ConvNext is the superior choice.
> > > >
> > > >
> > > >
> > > >
> > > > **Third**, the mIoU metrics are independent of the detectors，all BEV segmentation results are derived from the Cross-View Transformer (CVT), as utilized in both MagicDrive and BEVControl. We utilized the validation dataset generated by PerLDiff to evaluate BEVFormer, BEVFusion, and CVT. To enhance clarity and prevent misunderstandings, we will incorporate the following table in our revision.
> > > >
> > > > | Method                  | Detector  | FID↓      | mAP↑      | NDS↑      | mAOE↓    |
> > > > | ----------------------- | --------- | --------- | --------- | --------- | -------- |
> > > > | CLIP encoder            | BEVFormer | **12.77** | **25.41** | **36.26** | 0.73     |
> > > > | **PerLDiff (ConvNext)** |           | 13.36     | 25.10     | 36.24     | **0.72** |
> > > > | CLIP encoder            | BEVFusion | **12.77** | **15.80** | **24.45** | 0.81     |
> > > > | **PerLDiff (ConvNext)** |           | 13.36     | 15.24     | 24.05     | **0.78** |
> > > >
> > > > | Method                  | Segmentation | Road mIoU↑ | Vehicle mIoU↑ |
> > > > | ----------------------- | -------- | ---------- | ------------- |
> > > > | CLIP encoder            | CVT      | 60.49      | 27.06         |
> > > > | **PerLDiff (ConvNext)** |          | **61.26**  | **27.13**     |

---

### Author Response · Authors · 2024-11-22
**Response to All Reviewers by the Authors of PerLDiff**

We thank all reviewers for their time, insightful suggestions, and valuable comments. Below, we respond to each reviewer’s comments in detail and have revised the main paper and appendix according to their feedback. The main changes are listed as follows:

1. In Table 3, we have modified the entry to state that “Syn. \textit{val}\*” represents the synthetic \textit{validation} set generated by BEVControl, as identified by Reviewer sheB and gvD5.

2. In Section 4.2, we added an explanation and included a Fig. 14 in Appendix D that outlines why PerLDiff’s FID score is lower than that of BEVControl, in response to Reviewer sheB's question in Q2.

3. We clarified in Section 4.2 that we used the BEVFusion (Camera Only) version, addressing Reviewer sheB's question in Q2.

4. We provide a detailed explanation of the reasons for the performance gap in the KITTI dataset in Section 4.2, in response to Reviewer sheB's comment in Q3.

5. We added Fig. 16 in Appendix D to demonstrate the temporal continuity of generated images, in response to Reviewer gVD5's comment in W1.

6. We changed the term "BEV annotations" to "3D annotations" as identified by Reviewer 14qK.

7. We added Appendix C.4 titled "Effectiveness of Perl-based Cross Attention (Object)" as requested by Reviewer sheB in W2 and Reviewer 14qK in Q2.

8. We included Appendix C.5 titled "Effectiveness of Multiplication in Perl-based Cross Attention" as requested by Reviewer 14qK in Q3.

9. We added Appendix C.6 titled "Effectiveness of PerLDiff Based on ControlNet" as requested by Reviewer 14qK in Q4.

We hope that our efforts can address the reviewers’ concerns well. Thank you very much again!

Best regards,

Paper 52 Authors

---

### Author Response · Authors · 2024-11-25
**Request for Review Feedback**

Dear Reviewers,

As the deadline for the review process approaches, we eagerly anticipate your valuable feedback. We understand the various responsibilities you manage and are prepared to address any outstanding queries regarding our manuscript.

Please feel free to reach out if you require further information or clarification about our work. Thank you in advance for your time and dedication.

Best regards,

The Authors of Paper 52

---

### Author Response · Authors · 2024-11-26
**Feedback Request for Review Process**

Dear All Reviewers,

As we approach the review deadline, we would like to express our sincere gratitude for the time and expertise you have dedicated to evaluating our work. Your insightful comments and critiques have significantly contributed to our improvements, and we have worked diligently to address your concerns in our revised manuscript.

We are eager to receive your feedback on these revisions and are ready to clarify any remaining questions regarding our study. Please feel free to reach out if you need further information or have additional inquiries.

Thank you once again for your commitment to this process. We look forward to your valuable insights.

Best Regards,

Paper 52 Authors

---

### Meta-Review · Area_Chair_B9Gx · 2024-12-18

**Metareview:**

The paper presents "PerLDiff," a method designed to enhance control over the positioning of objects in street view synthesis. It ambitiously aims to outdo predecessors like BEVControl and MagicDrive by leveraging datasets such as NuScenes and KITTI for validation. Reviewers commend the paper for its clarity and articulate presentation, as well as the method's promising advancements in synthetic image generation control.

However, reviewers raised critical concerns regarding the novelty of the approach and its ability to generalize across different scenarios. Additionally, questions regarding the maintenance of temporal consistency and the specific choice of ConvNext for feature extraction were noted as potential issues.

In considering these points, the decision to recommend rejection is based on the paper's insufficient novelty and unresolved questions about its generalization capabilities. The additional concerns about the method's performance against more robust baselines solidify this stance.

**Additional Comments On Reviewer Discussion:**

During the review process, specific points were brought to the fore:

- Reviewer sheB questioned the method's effectiveness when compared to stronger baselines.
- Reviewer 14qK highlighted concerns regarding the novelty of the approach and its generalization capabilities.

The authors responded with further experiments and justifications for their choices, including the use of ConvNext and their approach to temporal consistency. While these responses partially addressed some of the issues raised by Reviewer sheB, they fell short of fully resolving the critical concerns of Reviewer 14qK regarding the novelty and generalization.

Ultimately, the recommendation is to reject the paper. Despite its clear presentation and the method's improved control in image synthesis, the paper does not adequately address the concerns about novelty and generalization. The additional issues raised regarding its effectiveness against stronger baselines further contribute to this decision. There are no ethical concerns to note in this recommendation.

---

### Decision · Program_Chairs · 2025-01-22

Reject